# A library of MiMICs allows tagging of genes and reversible, spatial and temporal knockdown of proteins in Drosophila

Sonal Nagarkar-Jaiswal[1], Pei-Tseng Lee[1], Megan E Campbell[1], Kuchuan Chen[2], Stephanie Anguiano-Zarate[1], Manuel Cantu Gutierrez[1], Theodore Busby[1], Wen-Wen Lin[1], Yuchun He[3], Karen L Schulze[3], Benjamin W Booth[4], Martha Evans-Holm[4], Koen JT Venken[5], Robert W Levis[6], Allan C Spradling[6], Roger A Hoskins[4], Hugo J Bellen[1,2,3,7,8]*

[1]Department of Molecular and Human Genetics, Baylor College of Medicine, Houston, United States; [2]Program in Developmental Biology, Baylor College of Medicine, Houston, United States; [3]Howard Hughes Medical Institute, Baylor College of Medicine, Houston, United States; [4]Life Sciences Division, Lawrence Berkeley National Laboratory, Berkeley, United States; [5]Verna and Marrs McLean Department of Biochemistry and Molecular Biology, Baylor College of Medicine, Houston, United States; [6]Department of Embryology, Howard Hughes Medical Institute, Carnegie Institution for Science, Baltimore, United States; [7]Jan and Dan Duncan Neurological Research Institute, Texas Children's Hospital, Houston, United States; [8]Department of Neuroscience, Baylor College of Medicine, Houston, United States

**Abstract** Here, we document a collection of ∼7434 MiMIC (Minos Mediated Integration Cassette) insertions of which 2854 are inserted in coding introns. They allowed us to create a library of 400 GFP-tagged genes. We show that 72% of internally tagged proteins are functional, and that more than 90% can be imaged in unfixed tissues. Moreover, the tagged mRNAs can be knocked down by RNAi against GFP (iGFPi), and the tagged proteins can be efficiently knocked down by deGradFP technology. The phenotypes associated with RNA and protein knockdown typically correspond to severe loss of function or null mutant phenotypes. Finally, we demonstrate reversible, spatial, and temporal knockdown of tagged proteins in larvae and adult flies. This new strategy and collection of strains allows unprecedented in vivo manipulations in flies for many genes. These strategies will likely extend to vertebrates.

*For correspondence: hbellen@bcm.edu

Competing interests: The authors declare that no competing interests exist.

## Introduction

Discoveries of gene function are most often driven by the integration of information related to cellular and subcellular protein expression patterns, genetic loss of function phenotypes, and protein–protein interactions. Antibodies to individual proteins can be used to localize their expression and for protein interaction studies. However, antibodies are not available for most proteins, and it is time consuming and expensive to generate them against a large number of individual proteins. Although antibodies have been documented against nearly 1900 fly proteins (*St Pierre et al., 2014*), most have been lost or are not obtainable anymore. An extensive survey reveals that antibodies are currently available against ∼450 different proteins in *Drosophila*, a mere ∼3% of all protein coding genes (*Adams et al., 2000*).

An alternative to generating antibodies to individual proteins is to tag the genes with a common protein segment for which antibodies are commercially available. Endogenous tagging of proteins

**eLife digest** In the last few decades, technical advances in altering the genes of organisms have led to many discoveries about how genes work. For example, it is now possible to add a specific DNA sequence to a gene so that the protein it makes will carry a 'tag' that enables us to track it in cells. One such tag is called green fluorescent protein (GFP) and it is often used to study other proteins in living cells because it produces green fluorescence that can be detected under a microscope.

It is labor intensive to add tags to individual genes, so this limits the number of proteins that can be studied in this way. In 2011, researchers developed a new method that can easily tag many genes in fruit flies. It makes use of small sections of DNA called transposons, which are able to move around the genome by 'cutting' themselves out of one location and 'pasting' themselves in somewhere else.

The researchers used a transposon called Minos, which is naturally found in fruit flies. When Minos inserts into a gene, it often disrupts the gene and stops it from working. However, the researchers could swap the inserted transposon for a gene encoding GFP by making use of a natural process that rearranges DNA in cells. This resulted in the protein encoded by the gene containing GFP and so it can be detected under a microscope. This method allowed the researchers to create a collection of fly lines that have the GFP tag on many different proteins.

Now, Nagarkar-Jaiswal et al. have greatly expanded this initial collection. More than 75% of GFP-tagged proteins worked normally and the flies producing these altered proteins remain healthy. It is possible to use a technique called RNA interference against the GFP to lower the production of the tagged proteins. Moreover, Nagarkar-Jaiswal et al. show that it is also possible to degrade the tagged proteins so that less protein is present. The removal of proteins is reversible and can be done in specific tissues during any phase in fly development. These techniques allow researchers to directly associate the loss of the protein with the consequences for the fly.

This collection of fruit fly lines is a useful resource that can help us understand how genes work. The method for tagging the proteins could also be modified to work in other animals.

with a fluorescent marker not only provides useful localization data, but may also permit the conditional removal of the gene products. Standard epitope tags such as Flag, HA, V5, GFP and other fluorescent tags allow the use of high-fidelity, commercially available antibodies against these tags, permitting biochemical experiments. Fluorescent tags also facilitate imaging in unfixed tissue or live animals. The most commonly used method of in vivo protein tagging is to introduce a transgene that contains a cDNA with a terminal epitope tag sequence that is expressed using an exogenous promoter (*Bischof et al., 2013*). Although this may permit the determination of the subcellular localization of the protein, it does not allow assessment of the native expression pattern. It may also affect the distribution and function of the protein, because the transgene is typically not expressed at the endogenous level. An alternative is to tag the genomic locus using a genomic transgene, rather than a cDNA, because native regulatory elements then control the expression patterns and levels (*Venken et al., 2008*; *Ejsmont et al., 2009*). This genomic transgene can be integrated by a site-specific integrase into a defined docking site to minimize chromosomal position effects on the expression of the transgene. The tagged genomic transgene allows the analysis of the spatial and temporal patterns of expression of the protein, but does not provide a means to alter the expression of the endogenous copy of the gene. Finally, the most effective and informative strategy is to tag genes in their endogenous locations (*Venken et al., 2011b*). A large collection of such GFP tagged genes would permit numerous powerful genetic and biochemical applications in addition to determining expression patterns and protein distributions. These include efficient immunoprecipitations with anti-GFP nanoantibody followed by mass spectroscopy (*Neumüller et al., 2012*), ChIP sequencing (*Nègre et al., 2011*), iGFPi (*Neumüller et al., 2012*) and deGradFP mediated protein degradation (*Caussinus et al., 2011*; *Urban et al., 2014*).

To tag numerous endogenous genes in *Drosophila*, protein trapping methods were previously developed based on screening untargeted insertions of transposable elements carrying a protein trap cassette. The first such protein trap vectors used *P*-elements, *piggyBacs* and *piggyBacs* with an internal *P*-element sequence (*Morin et al., 2001*; *Buszczak et al., 2007*; *Quiñones-Coello et al., 2007*; *Aleksic et al., 2009*). Only a small percentage of the transpositions function as protein traps, because

the insertion must be within the intron between two protein-coding exons, as well as being in the right orientation and reading frame to create an in-frame protein fusion when the artificial exon is spliced into the mRNA of the inserted gene. The pooled results of three major efforts (*Buszczak et al., 2007*; *Quiñones-Coello et al., 2007*; *Aleksic et al., 2009*) yielded less than 600 unique genes containing protein traps (*Quiñones-Coello et al., 2007*). Because of the insertion site biases of these transposons, it has been argued that this technology will not allow tagging more than about 5% of the *Drosophila* genes and that additional screening would yield very few novel tagged genes/proteins; hence, alternative approaches are needed (*Aleksic et al., 2009*).

We have previously shown that the *Minos* transposon-based MiMIC gene trap vector is much more efficient at generating intronic insertions in a much larger subset of *Drosophila* genes than either *P*-element or *piggyBac* vectors (*Venken et al., 2011a*). Moreover, MiMIC insertions in coding introns can be efficiently converted using RMCE to label the protein with a GFP or other epitope tag. We have vastly expanded the MiMIC collection, which now totals more than 7400 lines. We show that MiMIC is highly mutagenic and is an extremely efficient tool for gene/protein tagging. We created a new resource of 400 protein tagged genes and show that ~72–77% of essential genes with internal GFP tags are functional. Importantly, iGFPi and deGradFP permit a temperature-dependent conditional knockdown of gene function that mimics a severe loss of function in specific cells or tissues in most instances. Finally, we document the reversible tissue-specific knockdown of proteins and reversible loss of function of the *dunce* gene. Hence, the MiMIC protein trap collection is a valuable resource as it allows numerous different applications. The resource and tools described here will allow researchers to address important biological questions, particularly in adult flies, as very limited tools are available to conditionally remove and restore protein function in the adult.

## Results

### Expanding the MiMIC insertion collection

The goal of the *Drosophila* Gene Disruption Project (GDP) is to create resources to manipulate as many genes as possible (*Bellen et al., 2011*). Currently, we use a *Minos*-based transposable element, because *Minos* has less insertion bias than the *P*-element and *piggyBac* transposable elements (*Thibault et al., 2004*; *Metaxakis et al., 2005*; *Bellen et al., 2011*; *Spradling et al., 2011*). We previously engineered the MiMIC gene trap vector, which contains a phiC31 *attP* site, a splice acceptor (SA) followed by stop codons in the three reading frames, a polyadenylation signal sequence, the *yellow*[+] marker gene, and a second *attP* site in the opposite orientation (*Figure 1A*). We previously generated and sequenced 4464 insertion lines and reported a curated collection of 1269 MiMIC insertions (*Figure 1—figure supplement 1*, [*Venken et al., 2011a*]).

To expand the MiMIC collection, we generated and screened an additional 11,196 single-insertion lines, mapped 10,504 additional insertions to unique sites in the genome sequence using inverse PCR, and selected 6131 additional strains for the GDP collection. Consistent with previous studies of *Minos* insertion sites (*Metaxakis et al., 2005*; *Bellen et al., 2011*; *Venken et al., 2011a*), a very significant fraction of unselected insertions (38.6%) are in coding introns. As shown in *Figure 1B* we selected a total of 2854 MiMIC insertions in coding introns of 1862 distinct genes for inclusion in the GDP collection. Because many genes encode multiple protein isoforms, not all coding-intron insertions are equally useful. The collection includes 1732 insertions in constitutive coding introns that permit tagging of all annotated protein isoforms (Gold set), 814 insertions in alternative coding introns that permit tagging of more than 50% of annotated protein isoforms (Silver set), and 328 insertions in alternative coding that permit tagging of less than 50% of annotated protein isoforms (Bronze set). Note that 78 of the coding intron insertions map within coding introns of two distinct, overlapping genes. The expanded MiMIC collection also includes insertions in coding exons, untranslated regions, non-coding introns, and putative control regions (within 500 bp of the promoter) of 2860 protein-coding genes and 359 non-coding RNA genes, as well as 1439 intergenic insertions. In total, the collection comprises 7434 insertions in 7400 lines associated with 4367 genes; 34 lines contain two insertions each. The project website (http://flypush.imgen.bcm.tmc.edu/pscreen/) has a searchable database of all of the MiMIC lines that are currently available.

### MiMIC functions as an efficient gene trap

MiMIC insertions in coding introns that are in the same orientation as the transcript (~50% of coding intron insertions) should function as gene traps (GT). To further investigate the efficacy of MiMIC as

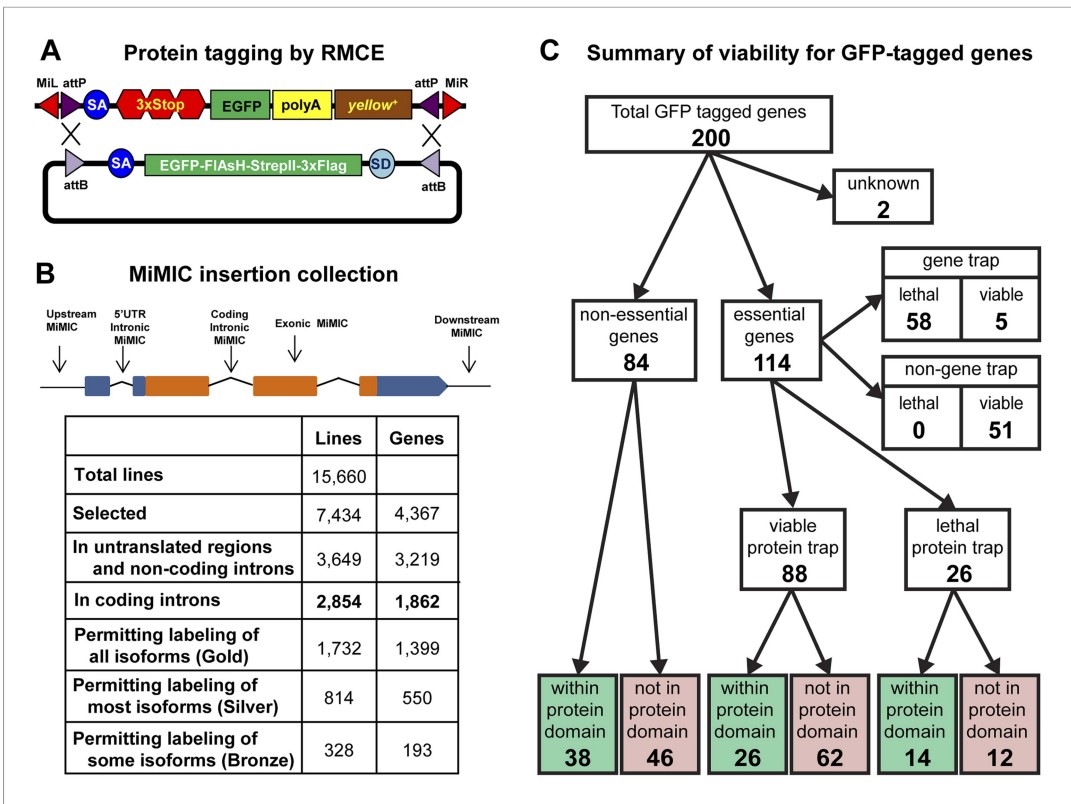

**Figure 1**. Protein tagging with the MiMIC system. (**A**) Schematic of Recombinase-Mediated Cassette Exchange (RMCE). The MiMIC transposable element consists of a splice acceptor (SA) followed by stop codons for all three reading frames, the EGFP coding sequence (a readout for stop codon skipping), a polyadenylation signal (PA) and the *yellow*+ marker flanked by two inverted *attP* sites and two *Minos* inverted repeats. The gene trap cassette between the two *attP* sites can be replaced with the protein trap cassette containing a splice acceptor (SA), EGFP-FlAsH-StrepII-TEV-3xFlag tag and splice donor (SD) flanked by two inverted *attB* sites. (**B**) Summary of relevant features of the MiMIC insertion collection based on the FlyBase 6.01 gene annotation. Note that the counts of insertions associated with features of the gene annotation do not sum to the total number of insertions. This is because some insertions are associated with more than one gene, some genes are associated with more than one insertion, and many genes have multiple annotated transcript isoforms. (**C**) Summary of viability of the sample of 200 lines with GFP-tagged genes.

The following source data and figure supplements are available for figure 1:

**Source data 1**. Characterization of 200 unique protein trap lines: MI: MiMIC insertion, GT: Gene Trap, PT: Protein Trap, Y/N: Yes/No and L/V: Lethal/Viable.

**Source data 2**. List of fly strains used in the study.

**Figure supplement 1**. Generating MiMIC insertions.

a gene trap and other important properties of tagged proteins/genes, we characterized 200 genes with intronic MiMIC insertions by tagging with RMCE. To establish whether MiMIC functions as an efficient gene trap when inserted in the proper orientation in a coding intron, and to assess the efficiency of the splice acceptor site (SA) we focused on lethality. Of the 200 genes analyzed, 114 are essential genes based on FlyBase records (*St Pierre et al., 2014*), 84 are non-essential, and 2 are unknown (*Figure 1C* and *Figure 1—source data 1*). The MiMIC insertions in coding introns of 63 of the 114 essential genes are inserted in the correct orientation to function as gene traps, and 58 of these 63 cause homozygous lethality and fail to complement null alleles or deficiencies (*Figure 1—source data 2*) that delete the target gene (*Figure 1—source data 1*); hence, MiMIC is

highly mutagenic. The remaining 51 essential genes have a MiMIC insertion in the non-GT orientation. All these lines are homozygous viable or complement a null allele or a deletion, showing that an intronic MiMIC inserted in the wrong orientation does not significantly disrupt gene function. Note that; 11/51 of the MiMIC-bearing chromosomes are homozygous lethal, but this lethality is caused by second site mutations based on complementation data (*Figure 1—source data 1*). In summary, our data show that MiMIC functions as designed in 109/114 insertion lines, and is thus a highly reliable gene trap. The data also show that the SA in MiMIC is effective and that the integrated artificial exon is probably not frequently skipped by the pre-mRNA splicing machinery.

## Tagging 400 genes with EGFP-FlAsH-StrepII-TEV-3xFlag

MiMIC insertions in coding introns can be used to introduce an artificial exon encoding one or more protein tags via RMCE as described previously (*Venken et al., 2011a*). We selected an EGFP-FlAsH-StrepII-TEV-3xFlag tag (hereafter abbreviated EGFP) that is flanked on either side with a 4X(GlyGlySer) flexible linker (*Figure 1A*). There are three versions of this protein trap cassette, one for each of the three intron reading frames. When the cassette with the proper reading frame is inserted into a coding intron in the proper orientation, the tag will be spliced into the mRNA of the target gene, and translation of this mRNA will result in a fusion protein with an internal tag inserted. We injected a plasmid DNA containing the donor tag cassette into embryos of ~700 MiMIC strains (intronic insertions) expressing phiC31 integrase. We identified $G_1$ progeny in which the RMCE event had occurred by the loss of the *y+* marker of the MiMIC gene trap cassette and determined which RMCE events were in the proper orientation using a PCR assay (*Venken et al., 2011a*). We established 450 independent stocks in which ~400 different genes are tagged. The success rate of obtaining at least one insertion in the correct orientation is currently 133/200 or 66%. Upon reinjection we obtained a 62% success rate. Hence, with two injections of about 500 embryos each we derived ~175/200 properly tagged genes.

## Internal tagging does not affect the function of ~75% of proteins

A concern with protein tagging is that the tag might disrupt protein function. We were not able to assess what fraction of proteins retains normal function when tagged at terminal or internal locations, despite an extensive survey of the current literature. As a proxy, we characterized tagged alleles of essential genes. If a tag disrupts the function of an essential protein, it should cause lethality. After tagging 114 essential genes using RMCE as described above, we performed complementation tests with null mutations or deletions (*Figure 1—source data 2*) to establish the fraction of tagged genes that failed to complement a severe loss of function or null allele. The lethality of 42 out of 58 lethal gene trap lines was reverted (72%) upon tagging. Moreover, all five of the essential tagged genes that contain an insertion in the GT orientation, but are viable, and 41 of the 51 essential tagged genes that contain an insertion in the non-GT orientation are homozygous viable or complement a severe allele or deficiency uncovering the gene (*Figure 1C*). Hence, 85 show no obvious phenotypes and three show subtle morphological phenotypes, suggesting a minimal impact on protein function. The remaining 26 tagged essential genes failed to complement null alleles or deficiencies. Hence, our data indicate that 72% (42/58) to 77% (88/114) of proteins tagged at various internal locations retain function or are not severely affected (*Figure 1C* and *Figure 1—source data 1*). However, a significant number of these chromosomes are homozygous lethal, and in almost all cases tested we have been able to eliminate the lethal mutation with backcrosses.

To assess whether internal tagging of proteins within annotated protein domains is more disruptive than tagging within other regions of proteins, we mapped the tag insertion site in each of 200 tagged proteins and determined its position with respect to annotated protein domains. We observe a strong relationship between tagging location, annotated protein domains, and retention of protein function. Of the 88 tagged essential genes that retain protein function, 26 tags are inserted within annotated protein domains and 62 are not inserted within an annotated protein domain ($\chi^2$, p = 0.0001). However, we found no such bias for internal tags that disrupt essential proteins: 14 tags inserted within annotated domains vs 12 inserted within unannotated sequences ($\chi^2$, p = 0.7). In summary, insertion of a tag within known, conserved protein domains is more likely to disrupt protein function, and inserting an artificial exon that encodes a tag is effective and not disruptive in 77% of the proteins tested.

## 90% of the tagged proteins can be detected in the third instar larval or adult CNS

The genes that were selected for tagging are a non-random set of genes whose expression is unknown in most cases. They were selected from requests made by members of the *Drosophila* neuroscience community. Although many were suspected to be expressed in the nervous system, quite a few are expressed in other tissues. As shown in *Figure 2A*, GFP expression can easily be detected in the brain (*Rab3 interacting molecule*: *Rim*), muscles (*Myosin Heavy Chain*: *MHC*), imaginal discs (*Abl tyrosine kinase*: *Abl*), salivary gland nuclei (*CrebA*), ovaries (*oo18 RNA-binding protein*: *orb*), and testis (*Syncrip*), in agreement with published expression data (*Lantz et al., 1994*; *Fogerty et al., 1999*; *Abrams and Andrew, 2005*; *Graf et al., 2012*; *McDermott et al., 2012*). Moreover, localization studies for Delta (Dl) and Ecdysone Receptor (EcR) show a precise colocalization of EGFP and the protein specific antibodies for Delta and EcR, respectively (*Figure 2—figure supplement 1*). Finally, proteins localize to the proper subcellular compartments as shown in *Figure 2B* for a cytoplasmic or organelle present in cytoplasm associated (*Catalase*), nuclear (*H6-like-homeobox*: *Hmx*) and membrane associated protein (*dpr15*) (*Figure 2B*) (*Beard and Holtzman, 1987*; *Hofmann et al., 2010*; *Özkan et al., 2013*). These examples indicate that the EGFP patterns faithfully report expression and protein distribution.

To determine what fraction of our set of 200 proteins are expressed in the CNS of third instar larvae, we performed immunostaining with anti-GFP antibodies and detected EGFP expression in 168/200 lines (84%). The lines that did not show detectable expression in larval CNS were tested for expression in adult brain, and EGFP expression was observed in 11/32 lines (6% of total analyzed lines). To further establish what fraction of the expression patterns could be monitored using unfixed tissue, we compared anti-GFP stained third instar larval brains with unfixed brains in 40 randomly selected lines. We were able to detect expression of EGFP fluorescence in unfixed brains in 38 of 40 lines. Although the expression patterns in each case generally matched the pattern observed by GFP antibody staining (*Figure 3*), the fluorescence intensity is often lower in unfixed tissue. Note that all the images presented in *Figure 2B* were taken at the same laser intensity and gain for each pair of images. Increasing the gain and laser intensity revealed very similar if not identical expression patterns. These data show that most internally tagged proteins are present at levels that permit imaging in unfixed tissue.

As part of our objective to generate a useful set of gene and protein trap lines for the fly community, we created a public website for the resource (http://flypush.imgen.bcm.tmc.edu/pscreen/rmce) containing all of the information about the MiMIC lines, insertion sites, associated genes, construct used for tagging, complementation data and images of brain expression patterns for each of the 200 lines described here (*Figure 3—figure supplement 1*). We are in the process of documenting the expression patterns of the remaining tagged lines and these data will be added to the online project database soon.

## iGFPi and deGradFP knockdown of tagged mRNAs and proteins

In addition to being very useful to determine expression pattern and subcellular localization of proteins, the GFP lines can be used to create conditional loss of function mutations via tag-mediated knockdown strategies. We tested two recently developed strategies to conditionally and reversibly knock down GFP tagged genes (iGFPi) (*Neumüller et al., 2012*) and proteins (deGradFP) (*Caussinus et al., 2011*) (*Figure 3A*).

There are three GFP RNAi transgenic lines available and each expresses a different short hairpin RNA (shRNA) against the EGFP tag under the control of the UAS/GAL4 system to knock down the expression of GFP-tagged genes (*Neumüller et al., 2012*). There are two important advantages of GFP RNAi over gene-specific RNAi: no off-target effects have been documented and a single set of highly selective and efficient GFP shRNAs can be used to knock down the expression of any tagged gene. By using specific GAL4 drivers, knock down of a gene in almost any tissue or stage in developing animals or adults can be achieved (*Jenett et al., 2012*; *Jory et al., 2012*; *Manning et al., 2012*). Alternatively, a GFP tagged protein can be degraded using the deGradFP system developed by *Caussinus et al. (2011)*. deGradFP is an ubiquitination-based system in which the F-Box domain of the Slmb protein N-terminus is fused to a single-chain nanoantibody fragment (vhhGFP4) that recognizes the GFP tag. The system uses the host ubiquitination machinery to target GFP-tagged proteins for proteasomal degradation. To compare these two knockdown strategies, we tested four EGFP tagged genes.

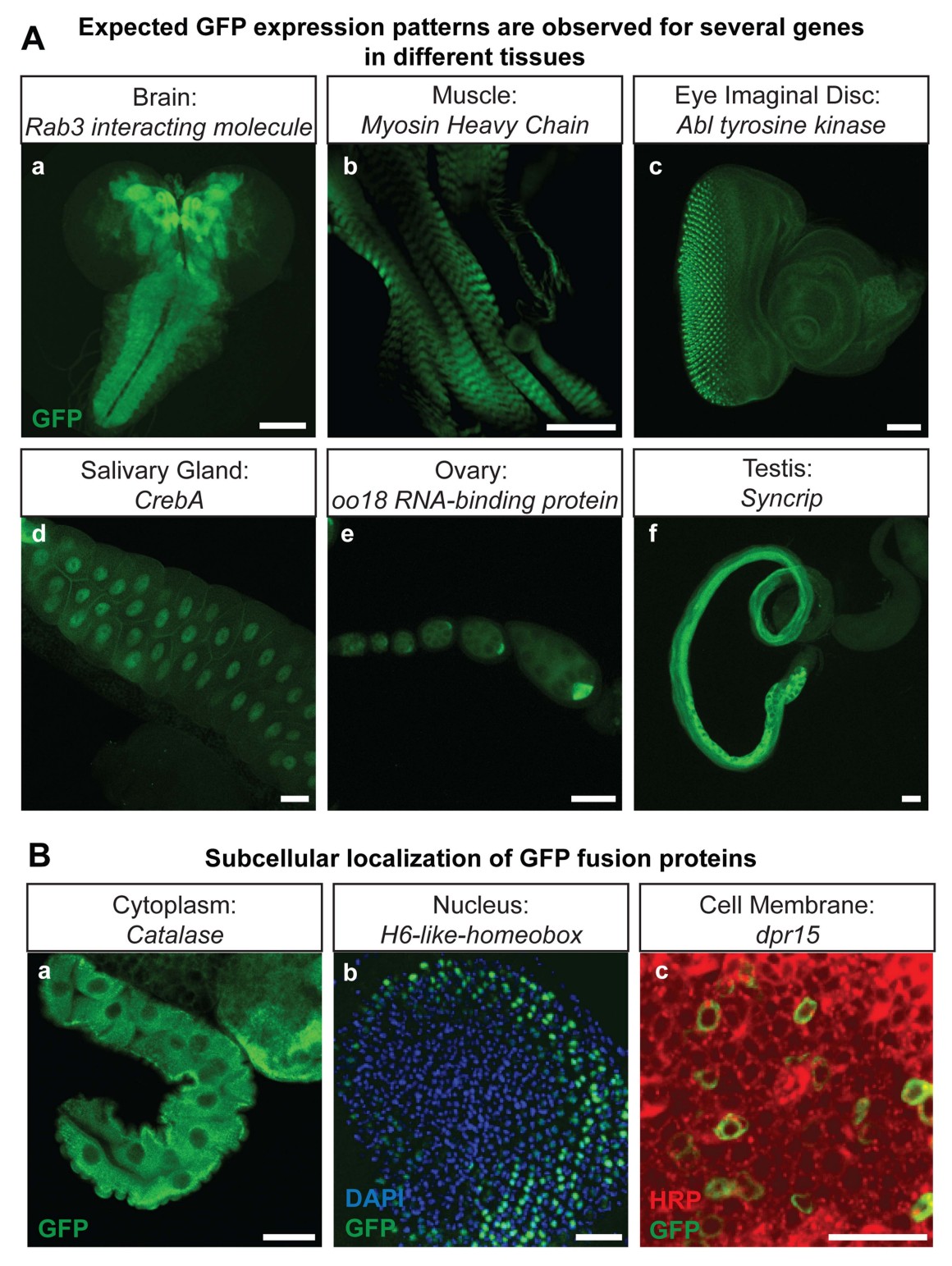

**Figure 2**. Protein expression analysis after RMCE. (**A**) Examples of GFP expression patterns in different tissues: (**a**) larval brain (*Rab3 interacting molecule*: *Rim*), (**b**) larval muscles (*Myosin Heavy Chain*: *MHC*), (**c**) larval eye imaginal disc (*Abl tyrosine kinase*: *Abl*), (**d**) larval salivary gland nuclei (*CrebA*), (**e**) adult ovaries (*oo18 RNA-binding protein*: *orb*), and (**f**) adult testis (*Syncrip*), were detected using anti GFP antibody. Scale bars, 50 μm. (**B**) Subcellular localization of GFP tagged proteins: (**a**) cytoplasmic/organelle associated localization of the enzyme Catalase in larval gut tissue, Scale bar, 100 μm (**b**) nuclear localization of H6-like-homeobox (Hmx) in eye imaginal disc (Green: Hmx-EGFP, Blue: DAPI) and (**c**) membrane localization of Dpr15 in larval brain

*Figure 2. continued on next page*

*Figure 2. Continued*
tissue (Green: Dpr15-EGFP, Red: HRP). Scale bars, 20 μm.
The following figure supplement is available for figure 2:

**Figure supplement 1**. Colocalization of protein trap GFP expression with specific corresponding antibodies.

## Knockdown of alpha-Catenin with iGFPi and deGradFP recapitulates known phenotypes

To test and compare knockdown strategies, we selected an intronic MiMIC insertion in *alpha-Catenin* (*α-Cat*). This protein plays a critical role at the plasma membrane where it acts as an essential physical linker between the cadherin-β-catenin complex and the actin cytoskeleton (*Desai et al., 2013*). We used RMCE to insert the EGFP tag (*Figure 3C*). The tag is not inserted in a known protein domain. The homozygous EGFP tagged gene is viable and displays no obvious phenotypes. Because *α-Cat* null mutants are embryonic lethal (*Sarpal et al., 2012*), and its loss specifically in the eye (*Seppa et al., 2008*) or ovaries (*Desai et al., 2013*) is associated with severe developmental or cellular defects, the tagged protein must be functional.

To determine the subcellular localization of the tagged α-Cat protein (α-Cat-EGFP-α-Cat), we performed co-immunostaining with anti-GFP antibody and anti-α-Cat antibodies (*Sarpal et al., 2012*) in the third instar larval eye disc. The tagged protein is ubiquitously expressed in the eye disc and localizes to adherens junctions as previously reported for α-Cat (*Sarpal et al., 2012*). The signals from each antibody colocalize and the tagged α-Cat protein localizes to the proper intracellular domain, similar to the wild type protein (*Figure 3D-a*; data not shown). As tagging with RMCE integrates an artificial exon, we determined whether exon skipping occurs. Given that MiMIC functions as an effective gene trap (see above) and since we used the same SA in the EGFP tagging construct (*Venken et al., 2011a*), we predicted that exon skipping would be rare. However, the integration of a splice donor (SD) in the inserted artificial exon cassette might alter the splicing properties and permit exon skipping. We performed Western blots of adult head extracts of *y w* and *y w; α-Cat-EGFP-α-Cat* with anti-GFP and anti-α-Cat antibodies (*Figure 4C*) (*Sarpal et al., 2012*). A single anti-α-Cat reaction band of 100 kDa is detected in *y w* flies, whereas *y w; α-Cat-EGFP-α-Cat* extracts show a single band of about 135 kDa, in agreement with the estimated molecular weight (MW) of 35 kDa for the protein tag (*Figure 4C*, left). The 135 kDa band was also observed in the anti-GFP immunoblot (*Figure 4C*, right). Hence, we are not able to detect untagged protein in homozygous tagged animals, indicating that the vast majority protein is tagged, but we cannot exclude that some protein in not tagged. This observation is important, as knockdown with iGFPi or deGradFP may be ineffective if exon skipping occurs.

Both iGFPi and deGradFP depend on the GAL4 binary system (*Brand et al., 1994*) to control spatial and temporal specificity. The GAL4 system exhibits some temperature dependency due to the presence of an *hsp70* promoter upstream of *GAL4* (*Duffy, 2002*). GAL4 is expressed at low levels at 18°C, while expression is elevated at 28°C. We tested this and show that with the *act-GAL4* driver, the temperature sensitive expression of GAL4 is quite pronounced (*Figure 4—figure supplement 1*). We therefore wondered whether we could use temperature to modulate the GFP tag-mediated knockdown efficiency. To test this, we ubiquitously expressed deGradFP using *act-GAL4* in flies expressing α-Cat-EGFP-α-Cat and maintained the cultures at 18°C or 28°C. Interestingly, *y w; α-Cat-EGFP-α-Cat* flies expressing deGradFP and kept at 18°C are viable, whereas animals kept at 28°C, to express deGradFP, are embryonic lethal (*Sarpal et al., 2012*).

To assess whether knockdown of α-Cat mediated by deGradFP or iGFPi phenocopies previously documented phenotypes, we expressed iGFPi or deGradFP using *ey-GAL4* in *y w; α-Cat-EGFP-α-Cat* and *y w* control flies. Animals were raised at 28°C to enhance the GAL4 expression level. Both knockdown experiments cause a rough eye phenotype: however iGFPi results in a more severe phenotype (*Figure 4D-b*). Knockdown with deGradFP causes rough eyes similar to one previously reported RNAi phenotype (*Seppa et al., 2008*) and knockdown using the α-Cat specific RNAi, FBst0033430 (*Figure 4—figure supplement 2-a*), but is less severe than other two unpublished RNAi phenotypes (FBst0038987 and FBst0038197), both of which are pupal lethal when raised at 28°C

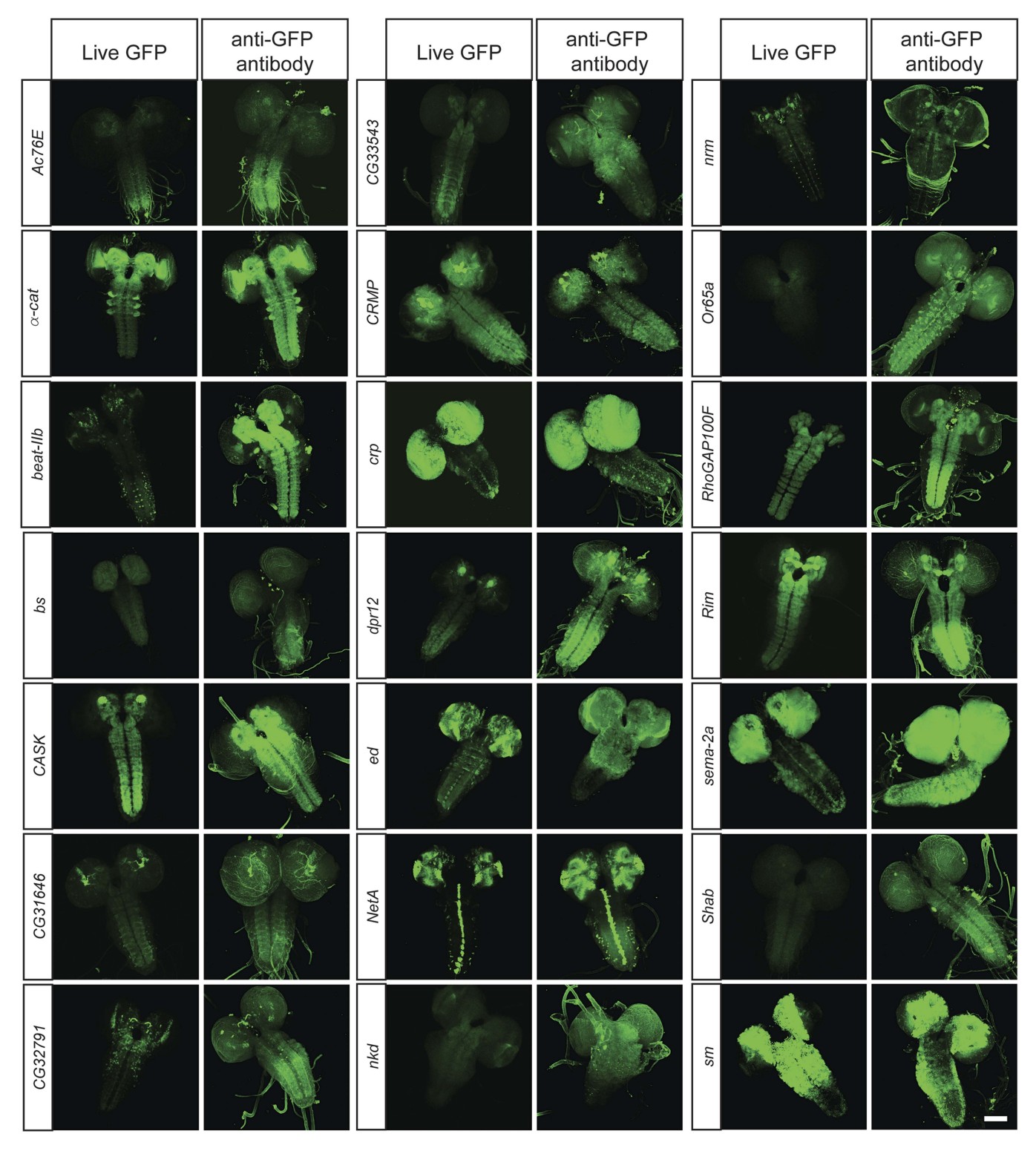

**Figure 3**. In vivo protein detection. Protein expression and distribution of GFP observed in unfixed third instar larval brains compared to those that were fixed and stained with an antibody against GFP. Each pair was imaged at the same confocal settings. Almost all pairs show very similar expression patterns but the gain or intensity needs to be adapted for genes that are expressed at low levels. Scale bar, 100 µm.

The following figure supplement is available for figure 3:

**Figure supplement 1**. A screenshot from the MiMIC protein database.

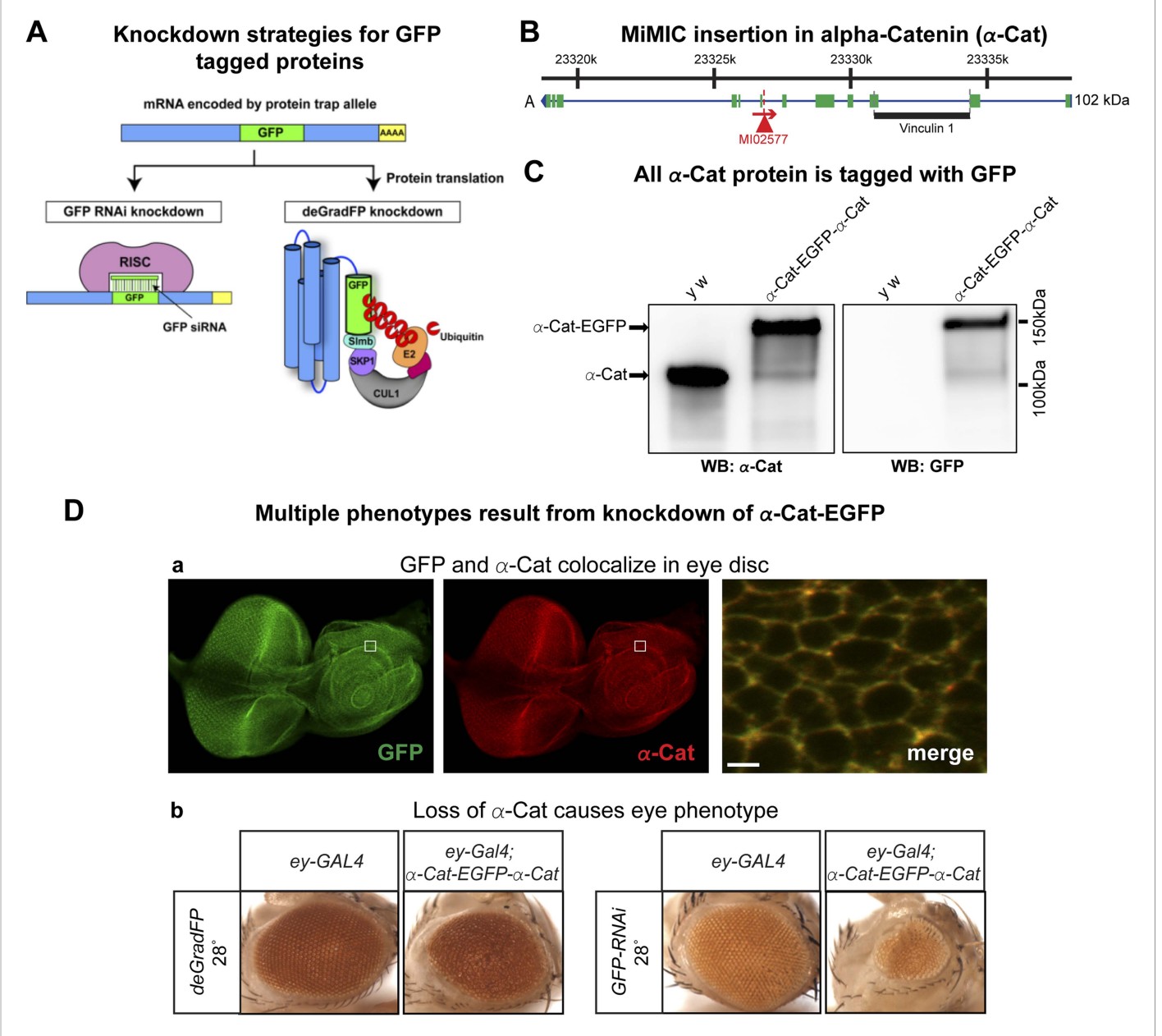

**Figure 4.** Tagging and knock-down of α−Catenin. (**A**) Two UAS/GAL4 based knockdown strategies targeting GFP-sequence containing mRNA or GFP fusion protein. Left: Expression of a GAL4-inducible shRNA transgene against GFP (UAS-GFP-RNAi) will result in gene knockdown by degrading mRNA of fusion protein; Right: GFP fusion proteins can be targeted for ubiquitination-mediated degradation by a modified ubiquitination system called deGradFP (*UAS-NSlmbvhhGFP4*). (**B**) Schematic diagram of *α-Cat* locus (based on FlyBase annotation release FB2014_05). The coding regions of tagged isoform are shown as green, 5′ and 3′ UTRs are in blue. The insertion site of MI02577 is shown with a red triangle and the orientation is shown with a red arrow. The black bar at the bottom of third exon represents the Vinculin 1 domain. (**C**) Western blots of adult head extracts from control *y w* and *y w; α-Cat-EGFP-α-Cat* were probed with anti α-Cat (on left) and anti GFP (on right). (**D**) Eye specific disruption of α-Cat expression causes eye related phenotypes. (**a**) An eye disc from a third instar larva expressing α-Cat-EGFP-α-Cat (*y w; α-Cat-EGFP-α-Cat*) is stained with anti GFP (green) and anti α-Cat (red) antibodies. Scale bar, 50 µm. On the right is a close up of the area boxed in eye disc on the left, which shows there is a strong co-localization of the GFP and α-Cat antibody signals. Scale bar, 5 µm. (**b**) Expression of deGradFP or GFP RNAi using *ey-GAL4* at 28˚C in the *α-Cat-EGFP-α-Cat* background result in rough eye phenotypes. Additionally, iGFPi knockdown causes a severe reduction in eye size.

The following figure supplements are available for figure 4:

**Figure supplement 1**. Temperature dependent Gal4 Expression.

**Figure supplement 2**. α-Cat knockdown with RNAi in developing eyes.

(*Figure 4—figure supplement 2-b–c*). Thus both tag-mediated knockdown strategies are effective at reducing gene function but, may not cause null phenotypes.

## Knockdown of discs large 1 (Dlg1) with deGradFP disrupts the maternal component

Null mutations in *discs large 1* (*dlg1*) cause larval lethality, oversized imaginal discs and brains in third instar larvae, and defects in neuromuscular junction development (*Stewart et al., 1972*; *Perrimon, 1988*; *Woods and Bryant, 1991*; *Woods et al., 1996*; *Budnik et al., 2006*; *Zhang et al., 2007*). Using RMCE, we converted the MiMIC line MI06353 into a protein trap allele (*Figure 5A*). The *dlg1* locus contains 16 long isoforms and five approximately 24 kDa isoforms, the latter of which may not encode functional proteins (*Mendoza-Topaz et al., 2008*). The MI06353 insertion is in an ideal location as it permits tagging of all the long, presumably functional Dlg1 isoforms. The EGFP insertion site is located between protein domains and is less likely to interfere with normal protein function (*Figure 5A*; *Figure 1C*). Homozygous *y w dlg1-EGFP-dlg1* flies are viable and do not exhibit obvious phenotypes, indicating that the tagged Dlg1 is able to function normally.

To assess whether exon skipping is occurring, we probed Western blots with anti Dlg1 antibody (raised against the second PDZ domain; see *Figure 4A*; [*Woods and Bryant, 1991*]) and noted a shift of ~35 kDa for the various long EGFP tagged Dlg1 isoforms (*Figure 5B*, top blot). Westerns with anti GFP show a nearly identical banding pattern in these flies, demonstrating that exon skipping is not occurring at a detectable level when assessed with western blots (*Figure 5B*, bottom blot).

Given that the previous GAL4 experiments indicate that there is significant temperature sensitivity we developed a simple diagram to explain how the temperature shifts are performed to temporally regulate expression of GAL4 and therefore deGradFP or iGFPi in subsequent experiments (*Figure 5C*). Note that development takes about twice as long at 18°C and hence embryonic development lasts for about 2 full days.

The Dlg1-EGFP-Dlg1 expression pattern and subcellular distribution as revealed with the anti Dlg1 and anti GFP antibodies also correspond to the published protein distribution (*Figure 4C-a–c*), further confirming that the GFP tag does not disrupt Dlg1 localization. The *y w dlg1-EGFP-dlg1;UAS-deGradFP/tub-GAL4* flies raised at 18°C develop normally and eclose without obvious phenotype. Moreover, GFP expression is not affected in third instar larvae (data not shown). In contrast, embryos that are raised continuously at 28°C are embryonic lethal, an earlier lethal phase than *dlg1* null mutants, which are third instar lethal. However, in zygotic null mutants, there is a large maternal component of Dlg1 (*Woods and Bryant, 1991*), and germ line clones of *dlg1* null alleles are embryonic lethal (*Perrimon, 1988*), suggesting that the maternally deposited protein is degraded in *y w dlg1-EGFP-dlg1; UAS-deGradFP/tub-GAL4* animals grown at 28°C. Furthermore, animals that are shifted to 28°C as either first or second instar larvae display hyperproliferative imaginal disc and brain phenotypes as third instar larvae (*Figure 5D–E*). In these animals, we also observe a significant decrease in Dlg1-EGFP-Dlg1 expression in the brain (*Figure 5D-c–e*) and in the wing disc (*Figure 5E*) compared with controls. We also detect a mild decrease in NMJ growth in terms of bouton number and synapse complexity in the temperature shifted animals compared to unshifted controls (*Figure 5D-a,d*-white arrows, data not shown), consistent with *dlg1* loss of function (*Zhang et al., 2007*). However, the decrease in Dlg1-EGFP-Dlg1 in the muscle is far less robust compared with other tissues, such as the brain, because the Tubulin driver leads to stronger expression in the brain compared to third instar larval muscles (data not shown). In addition to hyperproliferation, *y w dlg1-EGFP-dlg1; UAS-deGradFP/tub-GAL4* animals shifted to 28°C as first instar larvae, much like *dlg1* zygotic mutants, also have defects in cell morphology and organization as observed in the larval midgut compared to animals raised at 18°C (*Figure 5—figure supplement 1*). Taken together, these data show that deGradFP not only prevents zygotic expression of Dlg1, but also eliminates maternal deposited Dlg1, thus recapitulating phenotypes associated with zygotic null mutations and the lethality associated with mutant germ line clones.

## Knockdown of bruchpilot with iGFPi and deGradFP: differences and similarities

Bruchpilot (Brp) is a functional homolog of the human presynaptic protein ELKS/CAST/ERC (*Wagh et al., 2006*). It is essential for assembly of active zones, $Ca^{2+}$ channel clustering at NMJs, and

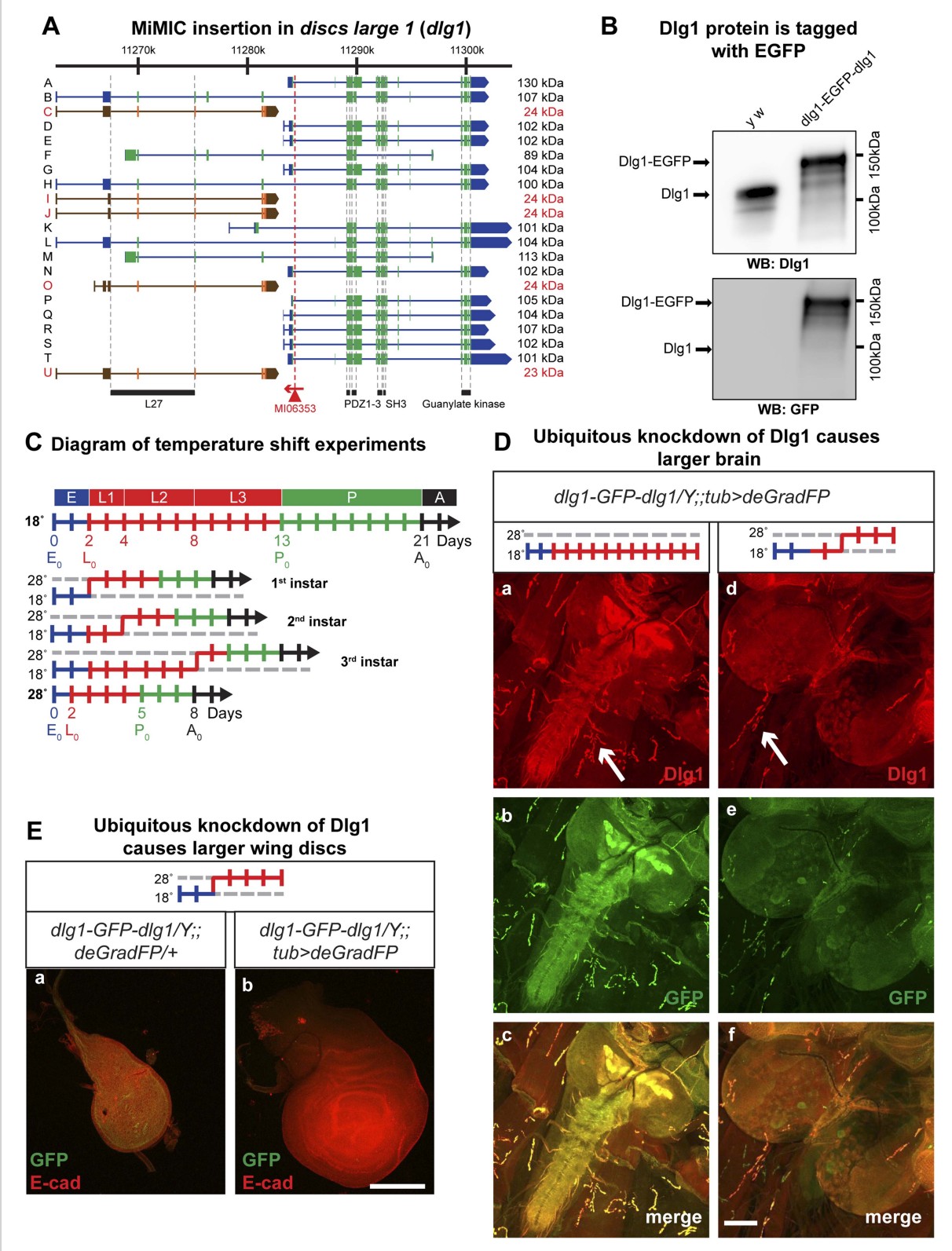

**Figure 5**. Knockdown of Dlg1-EGFP-Dlg1 with ubiquitously expressed deGradFP causes characteristic embryonic and larval phenotypes. (**A**) A schematic of the *discs-large1* (*dlg1*) gene region (based on FlyBase annotation release FB2014_05). The site of MI06353 insertion is shown with the red triangle. Each isoform is labeled by name and molecular weight of the protein product. Isoforms tagged with EGFP at the MiMIC insertion site have black labels, green

*Figure 5. continued on next page*

_Figure 5. Continued_

boxes (coding exons) and blue boxes (3′-, 5′-UTR exons); while isoforms not tagged with EGFP have red labels with orange and brown exon boxes. The black bars represent each protein domain as they map on the genetic sequence. (**B**) Western blots of head extracts from control: _y w_ and _y w dlg1-EGFP-dlg1_ were probed with anti Dlg1 (upper), which recognizes the second PDZ domain, and anti GFP (lower). (**C**) A diagram representing temperature conditions used in subsequent experiments to modify protein expression levels. The top bar indicates developmental stages: E (embryo), L1 (first instar larva), L2 (second instar larva), L3 (third instar larva), P (pupa), A (adult). The next bar is a time line (in days) of the developmental stages for animals kept at 18˚C. The bars below indicate the time at which the animals were shifted to 28˚C at the beginning of first, second or third instar, or kept continuously at 28˚C. (**D**) Third instar larval brains stained with Dlg1 (**a** and **d**) and GFP (**b** and **e**) antibodies. _y w dlg1-EGFP-dlg1;;UAS-NSlmbvhhGFP4/tub-GAL4_ animals that were raised continuously at 18˚C show robust larval brain expression of Dlg1 (**a**–**c**) with complete colocalization of Dlg1 and GFP expression (**c**). However, animals that were shifted from 18˚C to 28˚C as second instar larvae (see **C**) have less Dlg1 expression as judged by both Dlg1 (**d**) and GFP (**e**) antibody staining, however colocalization of Dlg1 and GFP is still present in some areas (**f**). Additionally, these brains (**d**–**f**) are significantly larger compared with controls (**a**–**c**), which is a characteristic phenotype associated with loss of function alleles of _dlg1_. The white arrows point to neuromuscular junctions. Scale bar, 100 μm. (**E**) Wing discs from third instar larvae labeled with GFP and E-cad antibodies. First instar larvae were shifted from 18˚C to 28˚C. The larvae that ubiquitously express deGradFP, _y w dlg1-EGFP-dlg1;;UAS-NSlmbvhhGFP4/tub-GAL4_ (**b**) have significantly larger wing discs compared with controls minimally or not expressing deGradFP, _y w dlg1-EGFP-dlg1;;UAS-NSlmbvhhGFP4/+_ (**a**), another traditional _dlg1_ mutant phenotype. Scale bar, 100 μm.
The following figure supplement is available for figure 5:

**Figure supplement 1**. Dlg1 knockdown results in aberrant cellular morphology and organization in larval gut.

release of neurotransmitters. _brp_ null mutants show loss of presynaptic dense projections and severe defects in synaptic transmission (_Kittel et al., 2006_). We created an EGFP tagged _brp_ allele using MI02987, which tags all but one annotated transcript (_Figure 6A_). The _y w; brp-EGFP-brp_ flies are homozygous viable and healthy, whereas severe loss of function _brp_ alleles cause third instar lethality (_Kittel et al., 2006_). To determine whether exon skipping occurs, we performed Western blotting on adult head extracts of _y w_ and _y w; brp-EGFP-brp_ animals with anti Brp (mAb nc82) (_Wagh et al., 2006_) and anti GFP antibodies The nc82 antibody recognizes two proteins of 165–175 and 200–210 kDa in _y w_ controls (_Figure 6B_, left) and two bands of 200–210 and 235–245 kDa in _brp-EGFP-brp_ flies (_Figure 6B_, right). These bands correspond to the 165–175 kDa and 200–210 kDa isoforms that are tagged with EGFP, as the western blots probed with anti GFP show two bands of 200–210 and 230–240 kDa. Hence, both isoforms are present in the tagged flies and we observe no evidence of exon skipping. The spatial protein distributions of the tagged isoforms are not affected in Brp-EGFP-Brp as a comparison of anti Brp and anti GFP staining patterns in the L3 brain and NMJs shows that the signals co-localize in the third instar neuropil and at the active zones in the NMJs, and the localization is similar or identical to published data (_Figure 6C_).

To compare the knockdown efficiencies in the adult eye, we expressed iGFPi and deGradFP using the _ey-GAL4_ driver in _y w; brp-GFP-brp_ flies and assessed knockdown efficiencies by western blotting of single adult head extracts using anti GFP antibody, which typically reveals only the abundant larger Brp-EGFP-Brp isoforms (upper band). _ey-GAL4_ drives expression in pupal and adult photoreceptors, lamina and medulla neurons, mushroom body neurons and other adult neurons (_Sheng et al., 1997_). Hence, a decrease of 50% in Brp-EGFP-Brp levels in whole head extracts must correspond to a very substantial loss of the protein in the cells where _ey-GAL4_ is expressed, which accounts for an estimated 40–60% of brain mass (_Figure 6D-a_). In agreement with the phenotype of a _brp_ severe loss of function mutation in photoreceptors, we observe a complete loss of on- and off-transients (_Figure 6D-b,c_) in electroretinograms (ERGs) using both iGFPi and deGradFP knockdown, indicating severe synaptic transmission defects of photoreceptors. We also observe a 50% reduction in ERG amplitude in both experiments, suggesting that Brp might play a role in eye development or the visual transduction pathway (_Figure 6D-b,c_). Similar reductions in ERG amplitude were previous described by _Wagh et al. (2006)_ using RNAi against _brp_ driven by _GMR-GAL4_, another eye-specific driver. We performed similar experiments using _GMR-GAL4_ to drive _UAS-deGradFP_ and _UAS-GFP-RNAi_ and observed virtually identical phenotypes to those described here for _ey-GAL4_ (data not shown). In summary, our data show that both knockdown systems are effective when the drivers are continuously expressed during development and in the adult.

To determine the time needed to remove the Brp-EGFP-Brp protein and cause synaptic transmission defects at third instar NMJ, we expressed deGradFP with _n-Syb-GAL4_, which drives expression in the

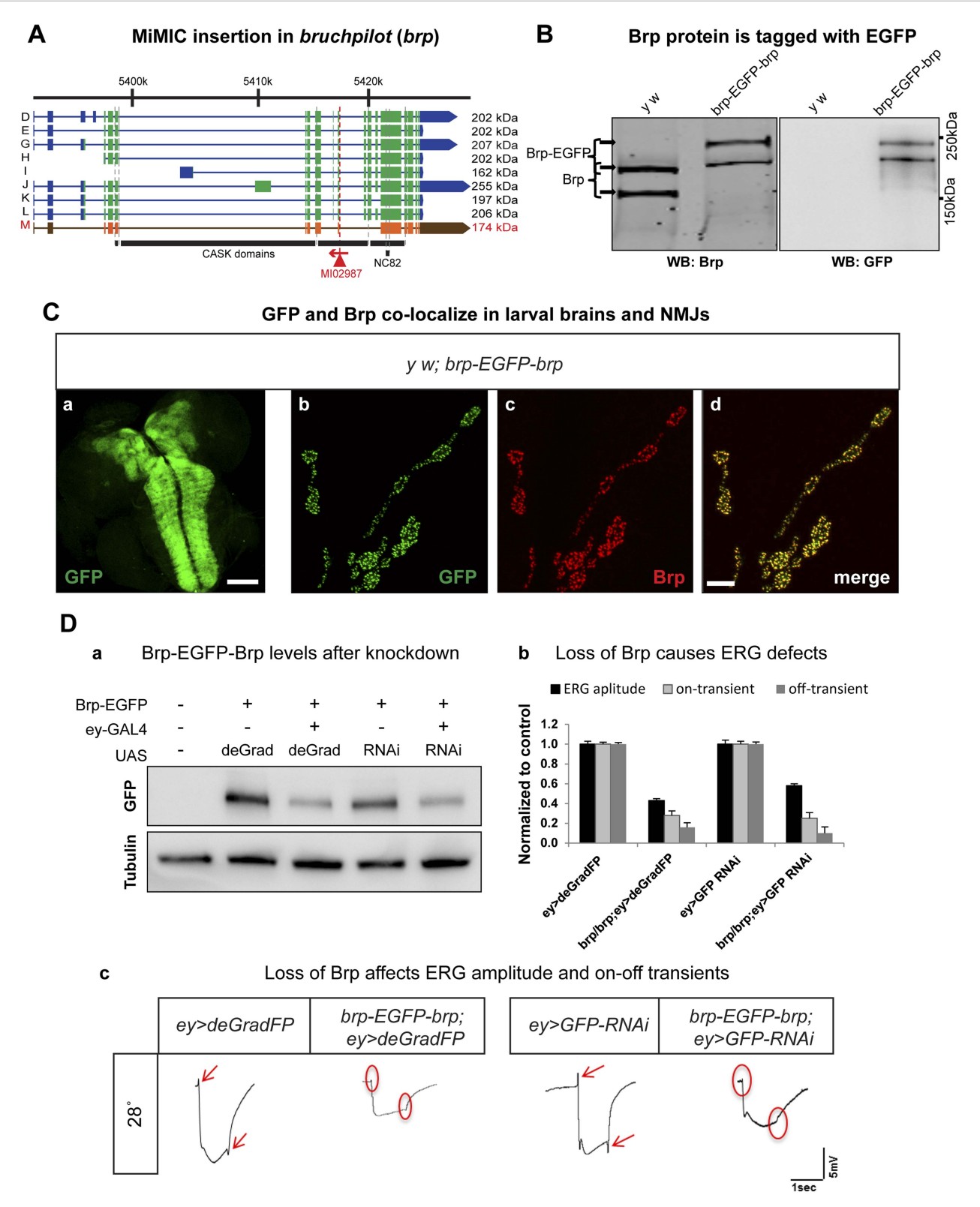

**Figure 6**. Conditional knockdown of Brp-EGFP-Brp phenocopies loss of function alleles. (**A**) A schematic of the *bruchpilot* (*brp*) locus (based on FlyBase release FB2014_05), coding exons in green and, 5′ and 3′ UTR exons are in blue. The insertion site of MI02987 is shown with a red triangle and the orientation is shown with a red arrow. On left, tagged isoforms are indicated in black and untagged isoform is indicated in red. The black bars represent

*Figure 6. continued on next page*

Figure 6. Continued

the CASK protein domains mapped onto genomic sequence. (**B**) Western blots of head extracts from *y w; brp-EGFP-brp* were probed with anti Brp (on left) and anti GFP (on right). (**C**) *y w; brp-EGFP-brp* third instar larval brain stained with antibody to GFP (**a**) and neuromuscular junction (NMJ) (**b–d**), was immunostained with antibodies to GFP (green) and Brp (red). Scale bars 50 μm and 7 μm, respectively. (**D**) Knockdown of Brp with deGradFP or iGFPi at 28°C using *ey-GAL4* driver results in altered physiology in the eye. (**a**) Western blot of adult head extracts probed with GFP antibody (and Tubulin as a loading control). Brp-EGFP-Brp levels are reduced when deGradFP or iGFPi are expressed using *ey-GAL4* driver (at 28°C). (**b**) Quantification of ERG amplitudes and on-and off-transients for each genotype shown in (**c**) (n = 6). ERG amplitudes and on- and off-transients were normalized with respect to controls. Error bars represent SD. (**c**) ERG traces of flies *ey-GAL4>deGradFP: y w;UAS-NSlmbvhhGFP4/ey-GAL4, brp-EGFP-brp; ey-GAL4>deGradFP*: *y w; brp-EGFP-brp;UAS-NSlmbvhhGFP4/ey-GAL4, ey-GAL4>GFP-RNAi: y w;UAS-GFP RNAi/ey-GAL4* and *brp-EGFP-brp; ey-GAL4>GFP-RNAi: y w; brp-EGFP-brp;UAS-GFP RNAi/ey-GAL4*. Normal on- and off- transients, as shown in the controls, are indicated by red arrows. When either deGradFP or iGFPi is expressed with *brp-EGFP-brp*, on- and off- transients are lost (indicated with the red circles) and ERG amplitude is reduced.

nervous system. Homozygous *y w; brp-EGFP-brp; deGradFP/n-Syb-GAL4* flies are viable and healthy at 18°C. We therefore raised *y w; brp-EGFP-brp; deGradFP/n-Syb-GAL4* larvae at 18°C until the early third instar larval stage and then shifted them to 28°C for 6–9 hr, 12–16 hr and 18–24 hr to assess the time needed to knock down Brp and cause a loss of synaptic transmission at the neuromuscular junction (*Figure 7A-a*). deGradFP knockdown efficiency is pronounced at NMJs in larvae that were kept at 28°C for 18–22 hr, as gauged by immunostaining with anti Brp (nc82) (*Figure 7A-b*) and anti GFP (data not shown).

To assess the physiological consequences of Brp loss, we performed electrophysiological recordings on third instar larval NMJs in 0.5 mM extracellular $Ca^{2+}$. We used *y w; deGradFP/n-Syb-GAL4* larvae with an untagged *brp* gene as control. There are no significant differences in excitatory junction potential (EJP) amplitudes and miniature junctional potentials (mEJPs) in controls (EJP: 21.9 ± 3.4 mV, and mEJP: 1.79 ± 0.18, n = 6) and 0 hr experimental animals (EJP: 22.9 ± 4.5 mV, and mEJP: 1.8 ± 0.13, n = 6) (*Figure 7A-c–f*). However when the temperature is shifted to 28°C during third instar larval development, there is a loss in amplitude of the EJPs that becomes progressively more severe the earlier that the shift is made. Larvae kept at 28°C for 18–24 hr have EJP amplitudes that are reduced by 76% (EJP: 5.5 ± 0.37 and mEJP: 1.87 mV ± 0.17, n = 5) compared to animals kept at 18°C (0 hr) (*Figure 7A-c–f*). Moreover, larvae kept at 18°C (0 hr) show no locomotion defects, whereas larvae kept at 28°C for 18–24 hr show severe locomotion defects (data not shown). In summary, an 18 hr period of deGradFP expression at 28°C during third larval instar development is sufficient to remove most of the Brp-EGFP-Brp protein and cause severe loss of function phenotypes.

To assess whether knockdown of Brp-EGFP-Brp affects adult behavior and to compare deGradFP and iGFPi efficiency in adult flies, animals expressing *n-Syb-GAL4* were shifted from 18°C to 28°C at progressively later stages (*Figure 7B*). The majority of animals raised continuously at 28°C and expressing deGradFP or iGFPi die as embryos or first instar larvae. Similarly, animals shifted to 28°C as first instar larvae (L1) die in late second instar (L2). However, later shifts show major differences between animals expressing deGradFP vs iGFPi: iGFPi animals shifted to 28°C as L2 larvae or adults show no obvious phenotypes, whereas deGradFP animals shifted as L2 larvae or adults show locomotor defects and die as pupae (L2 shifted) or adults (L2 or adult shifted). Moreover, temperature shifts to 28°C in adults allow flies to live for approximately 4 weeks after the shift. These data suggest that iGFPi is not as efficient as deGradFP at some stages of development or in some tissues, possibly because the Brp protein has a very long half-life and is less affected by depletion of mRNA levels by iGFPi than by degradation of the protein by deGradFP.

## Reversibly removing proteins

The temperature sensitivity of the GAL4-UAS system in combination with deGradFP should allow the reversible removal of proteins. A reversible knockdown has numerous advantages: it is quick, simple and only requires temperature shifts; it can be performed during development and in adult flies of any age; it allows the unambiguous assignment of the phenotype to the protein knockdown; it obviates the need of control genotypes; and it determines if damage to cells upon loss of a protein is reversible or permanent. We therefore tested if we were able to reversibly knock down and regenerate the proper expression pattern and protein distribution for a few tagged genes. We selected two proteins with a large extacellular domain, Roughest (*rst*) and NetrinA (*NetA*). The EGFP is inserted in the large extracellular domain of Rst and NetA. We were able to remove and re-establish protein expression and

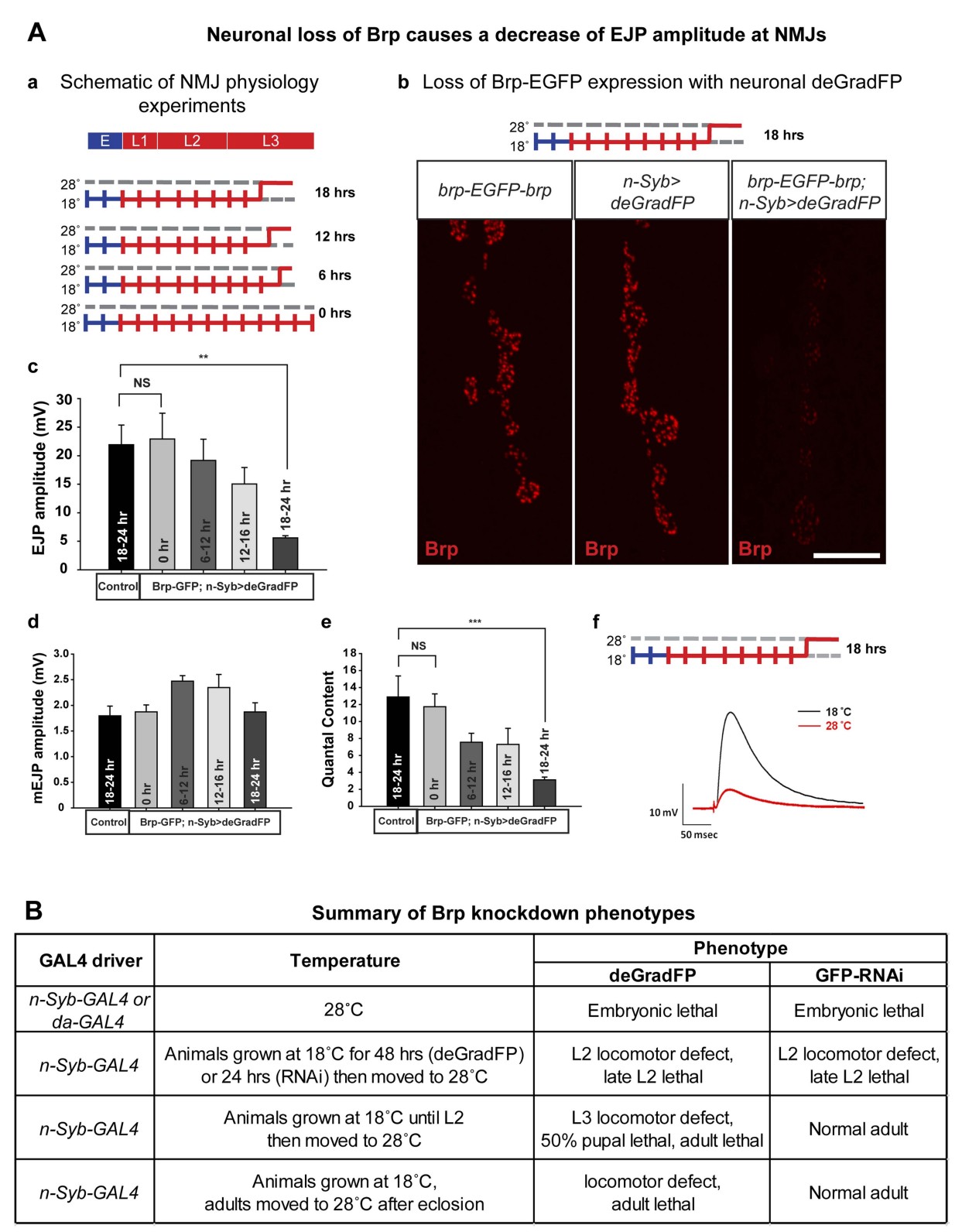

**A**  Neuronal loss of Brp causes a decrease of EJP amplitude at NMJs

**a**  Schematic of NMJ physiology experiments

**b**  Loss of Brp-EGFP expression with neuronal deGradFP

**B**  Summary of Brp knockdown phenotypes

| GAL4 driver | Temperature | Phenotype | |
|---|---|---|---|
| | | deGradFP | GFP-RNAi |
| n-Syb-GAL4 or da-GAL4 | 28°C | Embryonic lethal | Embryonic lethal |
| n-Syb-GAL4 | Animals grown at 18°C for 48 hrs (deGradFP) or 24 hrs (RNAi) then moved to 28°C | L2 locomotor defect, late L2 lethal | L2 locomotor defect, late L2 lethal |
| n-Syb-GAL4 | Animals grown at 18°C until L2 then moved to 28°C | L3 locomotor defect, 50% pupal lethal, adult lethal | Normal adult |
| n-Syb-GAL4 | Animals grown at 18°C, adults moved to 28°C after eclosion | locomotor defect, adult lethal | Normal adult |

**Figure 7**. Neuronal expression of deGradFP in *brp-EGFP-brp* flies causes defects in synaptic transmission. (**A**) Disruption of *brp* function with deGradFP at 28°C, using *n-Syb-GAL4* driver causes synaptic transmission defect. (**a**) Schematic diagram of temperature shift experimental parameters. *y w; brp-EGFP-brp;UAS-NSlmbvhhGFP4/n-Syb-GAL4* larvae were shifted to 28°C as Late L2 or L3 larvae for the time indicated on right. (**b**) NMJ6/7 from third instar larvae

*Figure 7. continued on next page*

Figure 7. Continued

that were raised at 18°C and shifted to 28°C for 18–22 hr were stained with an antibody to Brp (nc82). Brp expression is reduced in *y w; brp-EGFP-brp; n-Syb>deGradFP* compared with either *y w; brp-EGFP-brp* or *n-Syb>deGradFP*. Scale bar; 2 μm. (**c–f**) Electrophysiology was performed in *y w; brp-EGFP-brp; UAS-NSlmbvhhGFP4/n-Syb-GAL4* larvae that were shifted to 28°C at the time indicated in (**a**). EJP amplitudes (**c**), mEJP amplitudes (**d**), quantal content (**e**), of control and knockdown are measured. Both EJP amplitudes and quantal content in knockdowns show a ~76% reduction when larvae were raised at 18°C and shifted to 28°C for 18–22 hr. (**f**) Representative EJP traces obtained from controls (black) and 18–24 hr knockdowns (red). Each electrophysiology recording is performed at 0.2 Hz in 0.5 mM [Ca$^{2+}$] HL-3 solution. p value: **p < 0.01; ***p < 0.001 by Student's *t*-test. NS, not significant. Error bars indicate SEM. (**B**) A table summarizing lethality caused by disruption of Brp-EGFP-Brp with deGradFP or iGFPi using ubiquitous (*da-GAL4*) or neuronal (*n-Syb-GAL4*) GAL4 drivers after shifting animals to 28°C at different developmental stages.

proper distribution of Roughest and NetrinA in third instar larval brains (*Figure 8*). Both Rst and NetA have distinct expression patterns in the larval brain which are significantly reduced after 24 hr at the restrictive temperature, 28°C. However, when larvae are returned to the permissive temperature, 18°C, for 24 hr, protein expression is restored to a level comparable to larvae that were never exposed to the restrictive temperature (*Figure 8*). Therefore, through temperature manipulation, GAL4, and subsequently deGradFP expression can be controlled to reversibly disrupt target protein expression level. Given that GFP is in the extracellular domain, the degradation of these proteins must occur efficiently during protein synthesis.

Finally, to further assess the efficiency of protein knockdown through temperature regulation in combination with the deGradFP system, we preformed western blots against GFP for three other genes. Animals (*Frq1-EGFP-Frq1/Y;n-Syb-GAL4>deGradFP* and *CG14207-EGFP-CG14207/Y;act>GAL4* and *CG1632-EGFP-CG1632/Y;act>deGradFP*) kept at 18°C through eclosion were shifted as 1–3 day old adult animals to 28°C for either 1 or 3 days. We observe a range of protein loss with western blots against GFP from ~96% (Frq1-EGFP with n-Syb-GAL4) to 45% (CG1632-EGFP with act-GAL4) (*Figure 8—figure supplement 1*). Hence, knockdown efficiency is likely to vary significantly and should be assessed phenotypically and with western blots.

## Making flies *dunce* and smart again

To correlate the removal of a protein with a behavioral assay, we tagged the Dunce protein via RMCE using MI03415 (*Figure 9A*). *dunce* (*dnc*) encodes a cAMP phosphodiesterase (PDE) that is required for learning and memory (*Davis and Kiger, 1981*; *Qiu et al., 1991*). The EGFP cassette exon is positioned to tag all but one protein isoform (*Figure 9A*). The Dnc-EGFP-Dnc protein is expressed in the adult brain and is mostly restricted to mushroom bodies (MB), the center for learning and memory in *Drosophila* (*Figure 9B*). A more detailed analysis reveals that the fusion protein is expressed in all MB lobes (α, α′, β, β′ and γ) in the adult brain (*Figure 9B*), and the GFP staining reveals a much crisper expression pattern than the anti Dunce antibody (*Figure 9—figure supplement 1A*) (*Nighorn et al., 1991*). To knock down Dnc we used the *117y-GAL4* driver (*Armstrong et al., 1998*) which is expressed in MBs (*Figure 9—figure supplement 1*). The *dnc-EGFP-dnc;117y-GAL4/+;UAS-deGradFP/+* flies were raised at 18°C. 1 day after eclosion, the flies were transferred to 28°C for 3 days, and then returned to 18°C for 2 days. We observe a progressive loss of Dnc-EGFP-Dnc, and after 48 hr at 28°C we could not or barely detect the tagged protein (*Figure 9C*). Protein expression is restored 48 hr after a shift back to 18°C (*Figure 9C*). To determine whether the decrease in Dnc-EGFP-Dnc expression levels correlates with a known behavioral phenotype associated with loss of *dnc*, we performed the aversive olfactory learning assay (*Tully and Quinn, 1985*). As expected, the *dnc* mutant flies, *dnc*[1]/*dnc*[1], *dnc*[1]/*dnc*[ML] and *dnc*[1]/*dnc*[M14] (all partial loss of function combinations of *dnc* alleles) exhibit a 50% reduction in performance index when compared to *dnc*[1]/+, *y w*, Canton-S, and *dnc-EGFP-dnc* flies (*Figure 9D–E*). In contrast, *dnc-EGFP-dnc* flies expressing deGradFP under the control of the MB specific driver and maintained at 28°C for 3 days exhibit a performance index that is decreased by 70%, similar to the most severe *dunce* alleles (*Davis and Kiger, 1981*). The same flies were then returned to 18°C, Dnc-EGFP-Dnc expression in MB was re-established within 24 hr, and the learning phenotype was recovered fully within 48 hr. Hence, deGradFP is able to reversibly alter protein expression levels. In addition to reestablishing Dnc-EGFP-Dnc expression, the learning score is reestablished after 2 days at 18°C (*Figure 9F*). These data indicate that combining

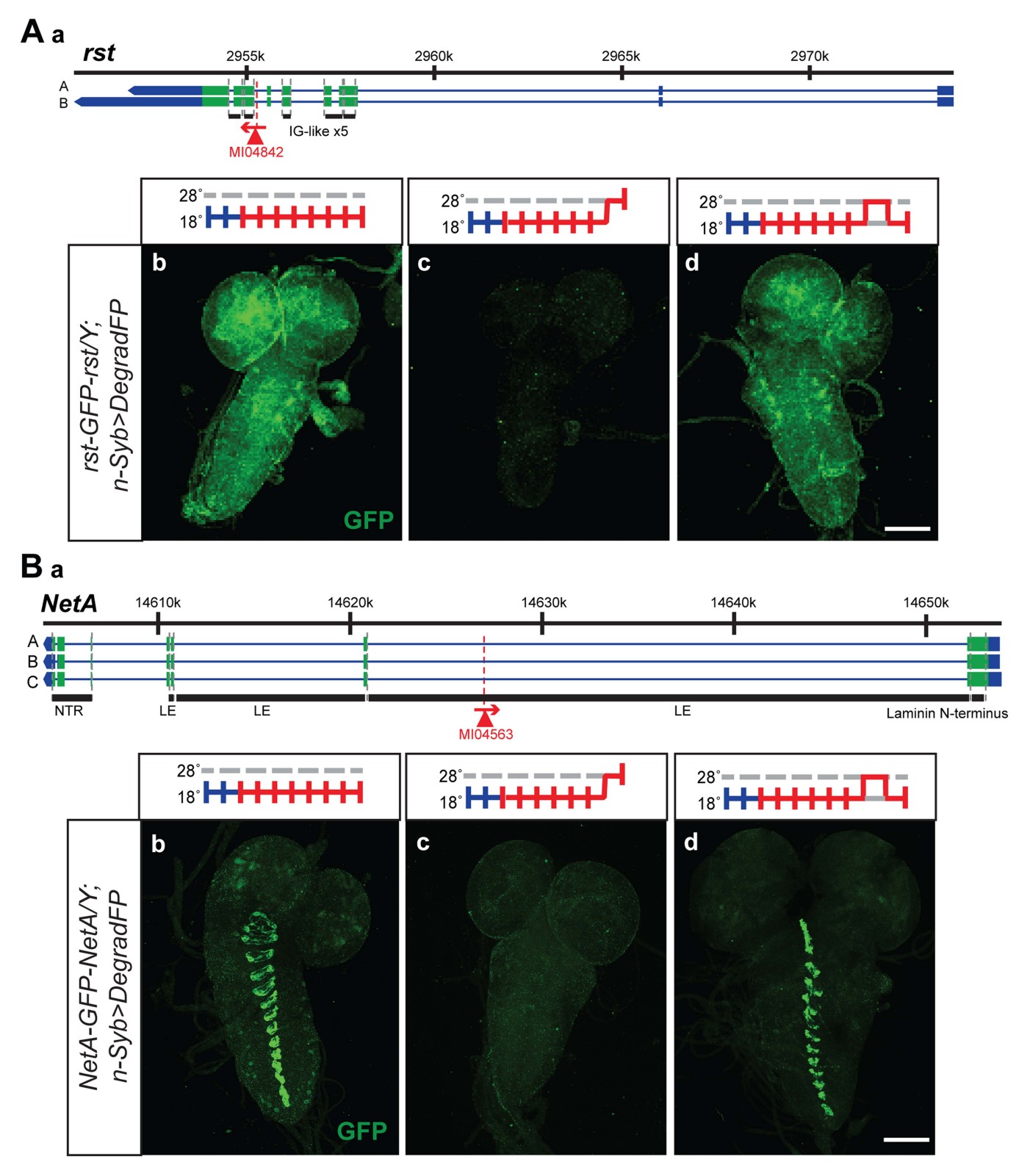

**Figure 8**. Knockdown and restoration of Rst and NetA protein expression in third instar larval brains. (**A**) (**a**) *Roughest* (*rst*) gene map (chromosome X, based on FlyBase release FB2014_05) with the position of the MI04842 insertion shown by the red triangle. (**b**) Expression pattern of Rst-GFP-Rst in third instar larval brain when raised constitutively at 18°C. Expression is barely detectable after animals have been shifted to 28°C for 24 hr (**c**). Expression is then restored by returning the animals to 18°C for 24 hr (**d**). (**B**) (**a**) *NetrinA* (*NetA*) gene map (chromosome X, based on FlyBase release FB2014_05) with the

*Figure 8. Continued*
position of the MI04563 insertion shown by the red triangle. (**b**) Expression pattern of NetA-GFP-NetA in third instar larval brain when raised constitutively at 18°C. Expression is barely detectable after animals have been shifted to 28°C for 24 hr (**c**). Expression is then restored by returning the animals to 18°C for 24 hr (**d**). Scale bars, 100 μm.
The following figure supplement is available for figure 8:

**Figure supplement 1**. Variable knockdown efficiency of deGradFP.

MiMIC-based EGFP tagging, *UAS-deGradFP* and *GAL4* drivers allows us to achieve not only spatial and temporal control of protein expression, but to do so in a reversible manner, in order to reduce and then restore gene function in live flies.

## Discussion

Here we describe a collection of 7434 MiMIC insertions that are inserted in or near ~4500 different genes. Of these, 2800 insertions are in a coding intron and can be used for protein tagging of more than 1862 different genes. We introduced an artificial exon that encodes a SA-Linker-EGFP-FlAsH-StrepII-TEV-3xFlag-Linker-SD tag via RMCE (*Venken et al., 2011a*) and created 450 protein trap alleles that tag 400 different genes. We present a detailed analysis of 200 tagged genes/proteins, and we performed a series of knockdown experiments on four tagged genes to assess the key features of this collection and demonstrate its power for functional analysis.

We previously described MiMIC insertions in a coding intron of three genes that had recessive lethal phenotypes and showed that the lethality could be reverted by removing the gene trap cassette by RMCE (*Venken et al., 2011a*). Our data show that 92% of MiMICs inserted in a coding intron in the gene trap configuration are highly mutagenic and function as gene traps, while 100% of those that are in the opposite configuration are not mutagenic. These data demonstrate that the selected SA (Splice Acceptor) in the MiMIC element, also used in the tagging cassette, is highly efficient and minimizes exon skipping. This is also corroborated by the fact that we only observe the predicted tagged isoforms in three genes that were tested by Western blotting: *α-Cat*, *dlg1*, and *brp*. This minimal level of exon skipping permits robust knockdown of the transcripts and proteins encoded by these genes using iGFPi and deGradFP (see below).

The analysis of the expression patterns and genetic tests of the 200 tagged genes reveals three highly valuable and surprising features of the library. First, EGFP expression can be detected in CNS of 90% of tagged genes. This is a surprisingly high number as it shows that most genes are expressed at sufficiently high levels to be detected with EGFP in their endogenous context. Second, intronic tagging does not obviously disrupt protein function in 77% of the 114 cases examined. This is also a surprisingly high portion of genes and is possibly due to the presence of unstructured linkers on either side of the tag. Third, expression of EGFP can be imaged in unfixed tissues in more than 90% of the samples tested. Note that this percentage has steadily risen during our project and the use of better confocal scopes has dramatically improved these numbers. The latter data are important as they document that most genes are expressed at levels that should permit direct imaging of GFP fluorescence in live tissue. These features make the collection very valuable and useful for many different applications.

Comparison of two knockdown strategies based on iGFPi and deGradFP using four different genes reveals some differences. Both strategies appear to be efficient during development. RNAi expressed in the zygote is unlikely to remove the maternal proteins deposited in the egg, but they are removed or strongly reduced by deGradFP in the case of *dlg1*. deGradFP is more efficient than the iGFPi system, except with knockdown of *α-Cat* in eyes where iGFPi exhibits a more robust phenotype. In contrast, for Brp-EGFP-Brp, RNAi knockdown causes similar phenotypes to deGradFP in embryogenesis and first instar larvae, but iGFPi leads to milder or no phenotypes at later stages, especially when the temperature shifts are performed in adults. These results suggest that the Brp protein is stable in adults and that the deGradFP system is able to target and disrupt Brp function, although it takes much longer to achieve than in embryos or larvae. In summary, both knockdown strategies should be explored when possible.

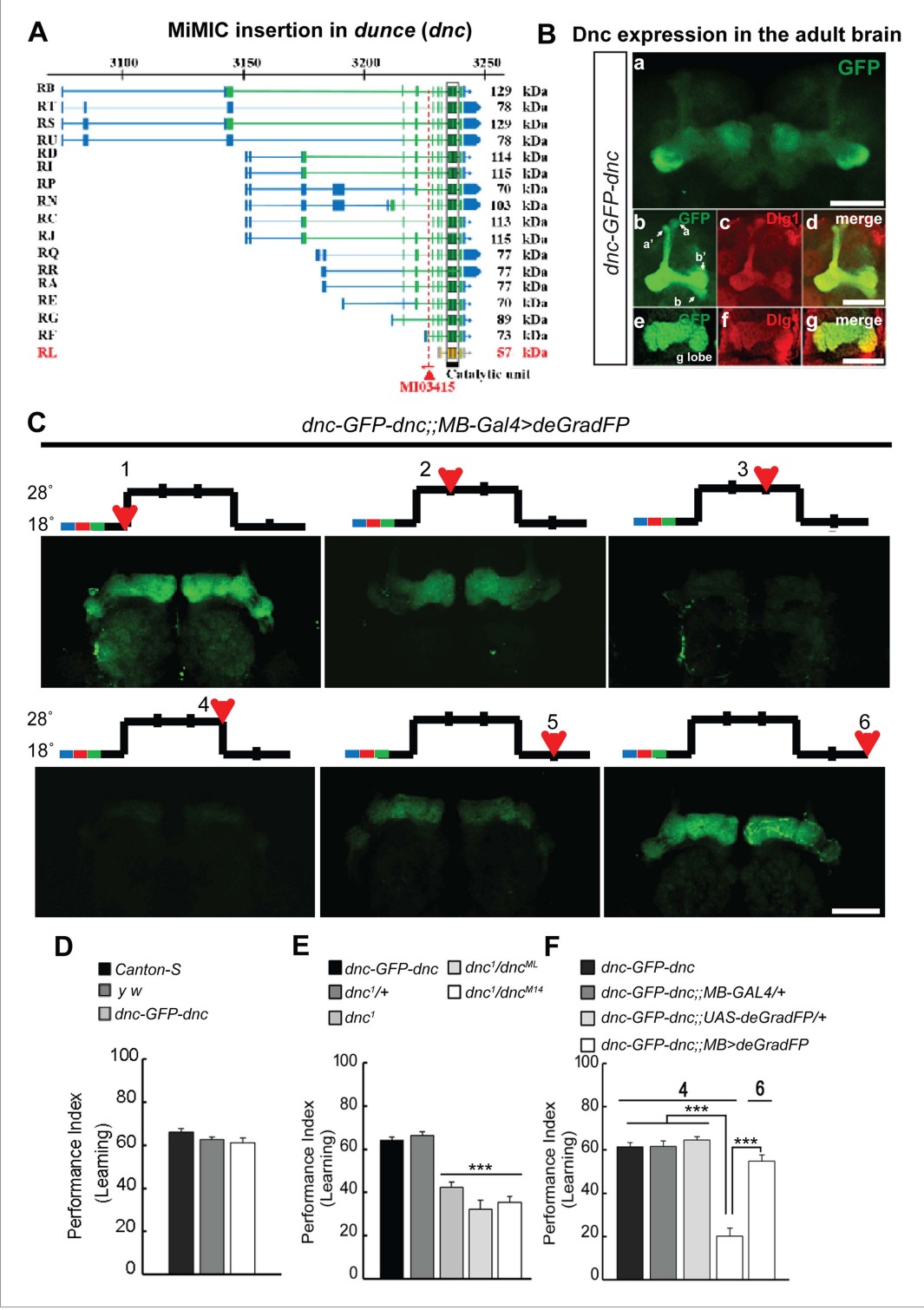

**Figure 9**. MiMIC mediated intronic tagging with EGFP permits a reversible spatial and temporal removal of proteins in flies. (**A**) *dunce* (*dnc*) gene map (chromosome X 3070.4 kb-3237.8 kb, based on FlyBase release FB2014_05) with the position of the MI03415 insertion shown by the red triangle. (**B**) *dnc-EGFP-dnc* expression pattern in adult brain (**a**). The α/β, α'/β' and γ lobes of mushroom body (MB) are shown below (**b–g**) stained with anti GFP (**b** and **e**) and anti Dlg1 (**c** and **f**) antibodies. (**C**) *dnc-EGFP-dnc* can be spatially and temporally knocked-down and re-expressed with temperature shifts modulating expression of *UAS-deGradFP* under the control of the MB-GAL4 driver, *117y-GAL4*. Adult brains are stained with anti GFP. (**D**) *dnc-EGFP-dnc* flies show a normal learning score similar to *Canton-S*

*Figure 9. Continued*

wild-type and *y w* flies. (**E**) Learning is impaired by temporal knockdown of *dnc* in MB caused by expression of *UAS-deGradFP* at 28°C for 3 days. (**F**) The learning deficit can be reversed with renewed *dnc* expression by shifting the animals back to 18°C for 2 days. The mean ± SEM is plotted for each treatment; n = 8 values for each group. ***p < 0.001. Scale bars, 50 µm.

The following figure supplement is available for figure 9:

**Figure supplement 1**. Dnc expression pattern in mushroom bodies in the adult head.

It was previously shown that during embryogenesis deGradFP requires less than 3 hr to remove a protein encoded by a transgenic *His2Av* construct (*Caussinus et al., 2011*). However, the temperature induced protein knockdowns that we observe take significantly longer. In the case of Brp-EGFP-Brp it takes at least 12 hr to observe a significant reduction in protein expression level in third instar larvae. Strong reductions in expression and a severe loss of synaptic transmission are observed 18 hr after a temperature shift in third instar larvae. For Dnc-EGFP-Dnc protein, we observe a severe knock down after 24 hr in adults, but it takes 48 hr at 28°C to observe a near complete loss of the protein in the mushroom bodies. In contrast to Brp-EFP-Brp and Dlg1-EGFP-Dlg1 where knockdown in adults can take up to 3 weeks or longer at 28°C to exhibit a severe phenotype. Yet, a ubiquitous adult knockdown with *actin-GAL4* driving UAS-deGradFP for α-Cat, WASp and Bifid (optomotorblind) results in lethality within a few days (data not shown). These observations suggest that differences in developmental stage, tissue, GAL4 strength and temperature will alter the dynamics of protein degradation with the deGradFP system leading to variation from protein to protein. As an additional note, we have observed that knockdown efficiency during developmental experiments is greatly enhanced when GAL4 is maternally deposited, by using GAL4 driver virgins.

The ability to tailor conditions will permit analyses of different knock down levels at different temperatures with different drivers. Given that allelic series have been shown to be extremely valuable in genetic studies, important biological information can be derived by varying the knockdown conditions, providing tremendous flexibility for in vivo analyses. While *Drosophila* is recognized as being one of the premier model organisms for the genetic analysis of gene function, one limitation has been the difficulty of conditionally knocking down genes or proteins in adults, especially in the adult nervous system. To achieve this goal, typically a complex set of reagents needs to be created, even with the simplest designs. These include a null or severe loss of function mutation, a UAS-cDNA rescue construct, a ubiquitously expressed GAL4 driver to rescue the mutation, and a GAL80$^{ts}$ construct to conditionally inactivate the GAL4 (*McGuire et al., 2003*). Using the library described here, we can now design large genetic screens in which we knock down proteins systematically in specific adult neurons or other tissues. The reversibility of the phenotype associated with the loss (28°C) and restoration (18°C) of the protein provide an important advantage as they permit us to directly pinpoint the cause of the phenotype, as well as the identity of the tissue that underlies the phenotype, as illustrated by the reversible MB-specific knockdown of the Dnc-EGFP-Dnc protein.

While we have focused on the uses of MiMIC insertions in coding introns in this paper, MiMIC insertions in other parts of genes and in intergenic regions can be used for many other genomic manipulation applications (*Wesolowska and Rong, 2010*; *Venken et al., 2011a*; *Venken and Bellen, 2012*; *Bateman et al., 2013*; *Chen et al., 2014*; *Vilain et al., 2014*). Moreover, MiMICS in coding introns have recently been used to create gene specific GAL4 lines that permit one to reveal where genes are expressed when endogenous protein levels are low (*Diao et al., 2014*, in press Cell Reports; *Gnerer et al., 2015*, in press Nucleic Acids Research). In summary, the resource that we have developed opens the door to numerous different applications and permits unprecedented manipulations of fly genes in vivo.

Given the breath of applications associated with in vivo tagged genes and proteins, a genome wide collection of MiMIC insertions and tagged genes would be extremely useful. We are therefore using RMCE to insert a GFP tag into an additional 1200 genes in which there is a MiMIC insertion in a coding intron. Finally, the GDP has recently embarked on a large-scale project to tag 5000 selected genes by inserting a MiMIC-like cassette that can be targeted precisely using CRISPR/Cas9 technology.

## Materials and methods

### Generation, mapping and selection of MiMIC insertion lines

New MiMIC insertions were generated by mobilizing the MI00827 insertion from the *X* chromosome to autosomal sites or by mobilizing the MI00000A or MI00000B insertions from a *TM3, Sb* balancer chromosome (*Figure 1—figure supplement 1*).

DNA segments flanking the new MiMIC insertions were amplified by inverse PCR, sequenced, and mapped by alignment to the reference genome sequence (*Venken et al., 2011a*). A detailed protocol is available on the GDP website (http://flypush.imgen.bcm.tmc.edu/pscreen/). Lines were selected for inclusion in the GDP collection by reviewing each insertion site with respect to annotated genes and the sites of other MiMIC insertions that were already part of the selected collection. Any insertion that was the first MiMIC hit in a gene was selected. When there were multiple insertions within a gene, priority was given to insertions within coding introns, coding exons, and introns in the 5′ UTR. Insertions within a coding intron or exon shared by the most annotated transcript isoforms were selected when possible. Multiple insertions were often selected for a gene if they hit different parts of the gene (e.g., one insertion in a coding intron and one in a coding exon) or if they provided a way of differentially tagging multiple annotated protein isoforms. We selected one insertion line in each of several tandemly repeated gene families (e.g., the histone and 5S rRNA genes), even though we could not localize the insertion to a unique copy within the tandem array. One of the goals of the MiMIC screen was to create an array of MiMIC insertions throughout the genome, spaced no more than 40 kb apart (*Venken et al., 2011a*). Toward this end, we selected intergenic MiMIC insertions if they were inserted >20 kb from other MiMICs already in the collection. The *attP* sites of intergenic MiMICs can facilitate targeted mutagenesis of nearby genes (*Wesolowska and Rong, 2010*, *2013*) and other genome engineering applications (*Venken and Bellen, 2012*; *Bateman et al., 2013*). Lines that were selected for the GDP collection were balanced, and their insertion sites were verified by resequencing, before delivery to the BDSC. The flanking sequences of all MiMIC insertions sent to the BDSC are available from the project website and have been submitted to GenBank.

### Protein tagging via RMCE

Approximately 150 males of a specific MiMIC line were crossed to 400 virgins females expressing ΦC31 integrase using the vasa promoter inserted on the X chromosome or on the IV chromosome (*y¹M{vas-int.B}ZH-2A w¹¹¹⁸; snaSco/SM6a* or *y¹M{vas-int.B}ZH-2A w\*; Sb/TM6b, Hu, Tb* or *FM7j, B[1]; M{vas-int.B}ZH-102D*). The resultant embryos were microinjected with pBS-KS-attB1-2-PT-SA-SD-[phase 0, 1 or 2]-EGFP-FlAsH-StrepII-TEV-3xFlag. The F0 flies were crossed to balancer virgins or males of *y¹w⁶⁷ᶜ²³; In(2LR)Gla, wgGla⁻¹/SM6a* or *y\* w\*; D/TM6b, Hu, Tb* or *FM7j, B[1]*. Transgenic F1 flies were scored for the loss of yellow+ (yellow− phenotype) and subsequently crossed to balancer virgins of *y¹w⁶⁷ᶜ²³; In(2LR)Gla, wgGla⁻¹/SM6a* or *y\* w\*; D/TM3, Sb, Tb*. Transgenic F2 flies were intercrossed to establish the final stock. Correct RMCE events were verified by PCR on genomic DNA obtained from 10–15 adult flies by using PureLink Genomic DNA Midi Kit (Invitrogen, Life Technologies, Grand Island, NY). PCR was performed using two tag specific primers Tag-F and Tag-R, and two MiMIC specific primers Orientation-MiLF and Orientation-MiLR, in four different combinations. First PCR reaction was performed with Orientation-MiL-F and Tag-R, a second PCR reaction with primers Orientation-MiL-F and Tag-F, a third PCR reaction with primers Orientation-MiL-R and Tag-R, and a fourth PCR reaction was performed with primers Orientation-MiL-R and Tag-F.

### Immunostaining: third instar larval brain, disc and NMJ

Third instar larvae were dissected for larval brains, imaginal discs, salivary gland, gut or NMJs in 1× PBS and fixed in 3.7% formaldehyde for 20 min (NMJ) or 30 min (brain, discs, salivary gland, gut) at room temperature and washed in 0.2% Triton X-100. They were then incubated for 1 hr at RT in 10% NGS-PBS-0.2% Triton X-100 and stained with primary antibodies diluted in 10% NGS-PBS-0.2% Triton X-100 for 2 hr at RT or overnight at 4°C. The samples were washed and incubated with secondary antibodies and HRP (where indicated) for 2 hr at RT. The samples were then washed, stained with DAPI (Invitrogen, Life Technologies, Grand Island, NY) for 20 min (where indicated) and mounted in Vectashield (Vector Labs, Burlingame, CA) and imaged with a Zeiss LSM710 or a Leica SP8 confocal microscope and processed using Adobe Photoshop (Adobe Systems Inc., San Jose, CA, USA).

## Immunostaining: adult brain

Whole-mount immunolabeling of the adult brain was performed as previously described (*Lee et al., 2011*). Briefly, brains were dissected in 1× PBS and fixed overnight in 4% paraformaldehyde in PBS on ice, transferred to 4% paraformaldehyde in PBS with 2% Triton X-100 at room temperature and vacuumed for 1 hr to remove the air sacs and left overnight in PBS with 2% Triton X-100 at 4°C. Following staining, brains were cleared and mounted in RapiClear (SunJin Lab Co., Taiwan) and imaged with a Zeiss LSM710 confocal microscope under a 20× or 40× C-Apochromat water immersion objective lens and processed using Adobe Photoshop (Adobe Systems Inc., San Jose, CA, USA).

## Antibodies used

Primary antibodies used: rabbit anti GFP 1:1000 (LifeTechologies A11122), guinea pig anti α-Cat 1:1000 (*Sarpal et al., 2012*), mouse anti Dlg1 1:500 (DSHB 4F3; [*Parnas et al., 2001*]), rat anti E-Cadherin 1:100 (DSHB DCAD2; [*Oda et al., 1994*]), mouse anti Brp 1:30 (DSHB nc82; [*Wagh et al., 2006*]), rabbit anti Dlg1 1:500 (Santa Cruz Biotechnology, Dallas, Texas), rabbit anti Dnc 1:10 (gift from Ron Davis), FITC conjugated rat anti GFP 1:500 (Santa Cruz Biotechnology, Dallas, Texas). Secondary antibodies used: Alexa 488 (Invitrogen, Life Technologies, Grand Island, NY), and Cy3 or Cy5 conjugated secondary antibodies (Jackson ImmunoResearch, West Grove, PA) were used at 1:500.

## Western blotting

Adult heads were homogenized in lysis buffer with an appropriate volume of 4 × Laemmli buffer and protease inhibitors. Samples were boiled at 95°C for 5 min and run on Mini-Protean TGX 4–20% gradient gels (cat#456-1034; Bio-Rad, Hercules, California) at 120 volts in a Bio-Rad Mini-PROTEIN NTM system followed by blotting onto a nitrocellulose membrane (Bio-Rad Trans-Blot Turbo RTA transfer kit) using the Trans-Blot Turbo Transfer system (Bio-Rad, Hercules, California). The membrane was blocked with blocking buffer (5% milk in PBS-Tw-0.2%) for 1 hr at room temperature and then probed with primary antibody diluted in 0.01% $NaN_3$ containing blocking buffer overnight at 4°C. After washing, the membrane was incubated with secondary antibody diluted in blocking buffer for 2 hr at room temperature and further developed using SuperSignal West Dura Extended Duration Substrate (#34075; Thermo Scientific, Waltham, MA) and Odyssey CLx Infrared Imaging System (LI-COR Biosciences, Lincoln, Nebraska). Signals were directly recorded and digitized by LAS 4000 machine (FUJIFILM Corporation, Europe). Images were processed using Adobe Photoshop (Adobe Systems Inc., San Jose, CA, USA).

## ERG (supplemental data/methods)

For ERG recording, *y*w*; brp-EGFP-brp; ey-GAL4* flies were crossed to *y*w*; brp-EGFP-brp; UAS-NSlmbvhhGFP4* and *y*w*; brp-EGFP-brp;UAS-GFP RNAi*. For controls *y*w*; ey-GAL4* flies were crossed to *y*w*; UAS-NSlmbvhhGFP4* and *y*w*; UAS-GFP RNAi*. Crosses were consistently kept at 28°C. Desired progeny *y*w*; brp-EGFP-brp; UAS-NSlmbvhhGFP4/ey-GAL4*, and *y*w*; brp-EGFP-brp; UAS-GFP RNAi/ey-GAL4*, and respective control *y*w*; UAS-NSlmbvhhGFP4/ey-GAL4*, and *y*w*; UAS-GFP RNAi/ey-GAL4* were collected and ERGs were performed as previously described (*Ly et al., 2008*). Briefly, adult flies were glued to a glass slide and a recording probe was placed on the surface of the eye, and a reference probe was inserted in the thorax. A flash of white light was given for 1 s, and the response was recorded and analyzed using AXON-pCLAMP 8 software.

## Electrophysiology

NMJ electrophysiology was performed as described previously (*Yao et al., 2009*). Briefly, wandering third instar larvae were dissected in ice-cold, 0.25 mM calcium HL-3 (70 mM NaCl, 5 mM KCl, 20 mM $MgCl_2$, 10 mM $NaHCO_3$, 115 mM sucrose, 5 mM trehalose, and 5 mM HEPES; pH 7.2), and rinsed with HL-3 containing 0.5 mM $Ca^{2+}$ concentration. The fillet was incubated in the latter solution for at least 3 min before recording. Recordings were made from body-wall muscles 6 (abdominal segment 3) with sharp electrodes filled with a 2:1 mixture of 2 M potassium acetate and 2 M potassium chloride. Data were collected only when resting membrane potential was below −65 mV. EJPs were evoked by directly stimulating the hemisegmental nerve through a glass capillary electrode at

0.2 Hz. Stimulus pulses were generated by pClamp 10 software (Molecular Devices, Sunnyvale, CA), and the applied currents were 6 μA ± 3 with fixed stimulus duration at 0.3 ms. 30 evoked EJPs were recorded from each muscle for analysis. Miniature EJP (mEJP) events were collected for 2 min. Both EJPs and mEJPs were amplified with an Axoclamp 900A amplifier (Molecular Devices) and digitized by Digidata 1550 Digitizer (Molecular Devices, Sunnyvale, CA). Experiments were performed at room temperature (20°C–22°C). EJPs were analyzed with pClamp 10, and mEJPs were analyzed using the Mini Analysis Program (Synaptosoft 29 Inc., Decatur, GA). The EJPs amplitudes were corrected by nonlinear summation (*Feeney et al., 1998*). The quantal content of evoked release was calculated from individual muscles by the ratio of the average EJP amplitude over the average mEJP amplitude.

## Learning assays

All flies are in *y w* background for conditional knock-down. For conditional protein knock-down, 1 day after eclosion flies are transferred from 18°C to 28°C for 3 days. For testing the reversibility, these flies are returned from the 28°C back to 18°C for 2 days. Aversive olfactory learning was performed by T-maze apparatus with a Pavlovian conditioning procedure as previously described (*Tully and Quinn, 1985*). Briefly, one training session consists of approximately 100 flies. During each session electrical shock is paired with the presence of one of the two odors (3-octanol and 4-methylcyclohexanol). Learning was measured 3 min after a single training session. Eight sessions were conducted for this odor-shock pairing (~800 flies) and then eight additional sessions were conducted with the other odor paired with the shock (1600 flies in total). The score is calculated as the number of flies avoiding the conditioned odor minus the number of flies avoiding the unconditioned odor divided by the total number of flies. The Performance Index (PI) is calculated as the average score of the two performances. Statistical analyses used KaleidaGraph 4.1 (Synergy Software, Reading, PA). Performance Indexes were evaluated via one-way ANOVA followed by planned comparisons among the relevant groups with a Tukey Honestly Significant Difference test. All data are presented as mean ± SEM. *p < 0.05.

## Acknowledgements

We thank Ying Fang for injecting hundreds of thousands of embryos; Jianping Li, Zhihua Wang, Qiaohong Gao and Lily Wang for creating the 15,000 MiMIC insertion stocks; and Hongling Pan for complementation tests. We thank Paolo Mangahas for his support. We thank Benjamin White, David Li-Kroeger and Shinya Yamamoto for comments on the manuscript. This research was funded by NIH/NIGMS R01GM067858 to HJB, 3R01GM067858-11S1 to SA, 3R01GM067858-09S1 to TB, and the Robert A and Renée E Belfer Family Foundation. KJTV was supported by startup funds kindly provided by Baylor College of Medicine and the McNair Medical Institute, and grants from the March of Dimes Foundation (#1-FY14-315), the Cancer Prevention and Research Institute of Texas (R1313), and the National Institutes of Health (1R21GM110190). Confocal microscopy was supported by NICHD P30HD024064 to the Baylor College of Medicine Intellectual and Developmental Disabilities Research Center. We thank the Bloomington Drosophila Stock Center (BDSC) for numerous stocks and the Developmental Studies Hybridoma Bank (DSHB) for antibodies. HJB is an Investigator of the Howard Hughes Medical Institute.

## Additional information

### Funding

| Funder | Grant reference | Author |
| --- | --- | --- |
| National Institute of General Medical Sciences (NIGMS) | | Sonal Nagarkar-Jaiswal, Pei-Tseng Lee, Megan E Campbell, Stephanie Anguiano-Zarate, Manuel Cantu Gutierrez, Theodore Busby, Wen-Wen Lin, Benjamin W Booth, Martha Evans-Holm, Robert W Levis, Allan C Spradling, Roger A Hoskins, Hugo J Bellen |

| Funder | Grant reference | Author |
|---|---|---|
| National Institute of Child Health and Human Development (NICHD) | R01GM067858 | Hugo J Bellen |
| March of Dimes Foundation | 3R01GM067858-11S1 | Stephanie Anguiano-Zarate |
| Cancer Prevention and Research Institute of Texas (CPRIT) | | Koen JT Venken |
| National Human Genome Research Institute (NHGRI) | | Koen JT Venken, Roger A Hoskins |
| National Institute of General Medical Sciences (NIGMS) | 3R01GM067858-09S1 | Theodore Busby |
| Baylor College of Medicine | McNair Startup Funds | Koen JT Venken |
| March of Dimes Foundation | #1-FY14-315 | Koen JT Venken |
| National Institutes of Health (NIH) | 1R21GM110190 | Koen JT Venken |

The funders had no role in study design, data collection and interpretation, or the decision to submit the work for publication.

## Author contributions

SN-J, P-TL, MEC, Conception and design, Acquisition of data, Analysis and interpretation of data, Drafting or revising the article; KC, Performed the electrophysiology assays, Acquisition of data, Analysis and interpretation of data, Drafting or revising the article; SA-Z, MCG, TB, W-WL, YH, ME-H, RWL, Acquisition of data, Analysis and interpretation of data, Drafting or revising the article; KLS, BWB, ACS, RAH, Analysis and interpretation of data, Drafting or revising the article; KJTV, Conception and design, Drafting or revising the article; HJB, Conception and design, Analysis and interpretation of data, Drafting or revising the article

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
