## [Decision Letter]

Thank you for sending your work entitled “A library of MiMICs allows intronic EGFP tagging for reversible spatial and temporal knockdown of proteins in *Drosophila*” for consideration at *eLife*. Your article has been favorably evaluated by K VijayRaghavan (Senior editor), Mani Ramaswami (Reviewing editor), and two reviewers.

The Reviewing editor and the reviewers discussed their comments before we reached this decision, and the Reviewing editor has assembled the following comments to help you prepare a revised submission. Should this be accepted for publication, we would like to consider this as a Tools and resources article, a new category we are planning to introduce.

All the evaluators see the importance and interest of your manuscript. However, all concur that the comments below are useful and relevant to better establish and communicate the main conclusions and value of the manuscript. You should therefore address these in a revised manuscript. If there are any that you find to be difficult or feel are irrelevant, please let us know indicating your reasons and we will consider your arguments.

This manuscript represents a beautifully presented description of a new library of MiMIC insertions that will be of great benefit to the *Drosophila* community. This study by Bellen and colleagues describes an expanded resource of MiMIC insertions that can be of multiple use to the fly community. Technologically, it is a follow-up from earlier work (Venken, 2011, Nat Met) using the same technology, however providing lager numbers and showing additional applications. As a proof of principle, the authors tag 400 genes with a EGFP tag and provide evidence that incorporation of the tag generally does not interfere with protein function. The authors then show how the EGFP tag can be used with GFP-RNAi and deGrad-FP to knockdown tagged mRNAs and proteins. Lastly, they demonstrate the utility of the MiMIC lines by providing three examples of how these reagents can be used to genetically dissect gene function in the nervous system.

The major impact of this study is the numbers and not the biology. The study isolated 6,800 new MiMIC lines, 1,733 are in “coding” introns, of which 1,658 are predicted to label all isoforms. A selected set of lines was used to tag 200 genes with GFP by RMCE and a few of them were investigated in a bit more detail in this study. The authors state that 77% of the internally GFP tagged proteins are functional and >90% can be detected in live tissue. The authors conclude that their tagging generally enables tissues specific knock-down, which can be controlled in time using deGradFP or GFP-RNAi.

These conclusions are in principle valid and important, however there are several concerns:

1) The authors conclude that 77% of the tagged lines are functional as 88 of 114 protein traps in (predicted) essential genes are viable. In the same paper they state that only 58 of 63 gene traps (within the same gene set of 114 genes) are actually lethal. Thus, not all gene traps in “essential” genes are lethal and hence the authors' conclusion of 77% is not valid. They should rather only count how many of the 58 lethal gene traps were reverted to viable proteins traps when inserting the GFP cassette. That number should be more accurate and takes potential exon skipping into consideration.

More generally, the conclusion that 77% of the tagged proteins “retain function or are not severely affected” is based, at least in part, on the assumption that their EGFP cassette sequence is incorporated into nearly 100% of the mature message. However, sequence context could potentially affect the efficiency of cassette splicing. The authors should consider toning down their language.

For example, in the second paragraph of the subsection headed “Knockdown of alpha-Catenin with iGFPi and deGradFP recapitulates known phenotypes”, in the Results section, the authors state: “Hence, we are not able to detect untagged protein in homozygous tagged animals, indicating that the vast majority or all protein is tagged.” However, a small amount of endogenous protein appears visible on the presented *alpha-Cat*, Dlg1 and Brp westerns. Therefore the authors should rewrite this sentence. Likewise, in the same paragraph, when stating that “westerns with anti-GFP show a nearly identical banding pattern in these flies, demonstrating that little or no exon skipping is occurring (Figure 4, bottom), the authors should delete “or no”. Addressing such caveats throughout the paper and in the discussion would in no way diminish the usefulness of this resource.

2) The beauty of a GFP tagged protein is to investigate its sub cellular localisation. However, the authors only provide tissue expression data and that only from the larval nervous system. This information does not go much beyond the FlyAtlas data of gene expression. Not even the nuclear factors like Jumu-GFP or H6-like homeobox-GFP are shown to localize in the nuclei. Were any new or unexpected patterns discovered? The authors state that Ac78c-GFP labels a subset of neuronal membranes. I cannot see this from the data presented. A few images particularly carefully comparing antibody and tagged-GFP labelling will be useful.

3) Figure 2 is redundant with Figure 2. It would be much more useful to give an overview how the 200 genes for expression analysis were selected from the 400 tagged genes, which again were selected from the 1,600 MiMICs. Can the authors expand their explanation for how these lines were chosen that is a bit more specific than “non-random”? Why were they “suspected” to be expressed in the nervous system? Have they been selected using fly atlas above a certain RNA expression level?

4) The paper mainly focuses on the larval nervous system. As this resource is supposed to be of general use, it would be useful to show that the used muscle myosin heavy chain splice acceptor is indeed efficiently used in other tissues like muscle, epidermis, endoderm etc. This could be easily shown by selecting a couple of ubiquitous essential genes that should result in lethality when GFP RNAi is driven in various tissues.

5) The authors nicely show that *α-CatGFP* overlaps with anti-α-Cat antibody in eye discs and that deGradFP and GFP RNAi driven by ey-GAL4 can result in some eye phenotypes in these flies. How do these phenotypes compare to the various available *α-Cat* RNAi lines? How strong is the protein reduction? As western blots of the Cat-GFP lines are shown, these data should be easy to acquire. It would be even more important to generalize knock-down efficiency by testing a few different number used genes with western blots. Again, these are straight forward experiments that would be very valuable for the community to judge, how well the technology works in general and not only in a few selected examples.

6) The authors argue that one major advantage of their technology is to eliminate gene function tissue specifically. Following this logic, why was Tub-GAL4 used to knock-down Dlg1-GFP in Figure 4? The loss of epithelial polarity in Figure 4 vs. g is not clear. Better data need to be provided supporting that conclusion.

7) We do not understand the interpretation of the Figure 5. anti-Brp clearly shows two bands on western from wild type but only one from Brp-GFP flies. However, anti-GFP detects two bands from Brp-GFP with inverted intensities. The authors interpret these data such that anti Brp does not recognize some Brp-GFP isoforms, because of the GFP insertion. My interpretation is that rather splicing is abnormal, inverting the intensities and partially skipping the exon that contains the anti-Brp epitope.

---

## [Author Response]

*1) The authors conclude that 77% of the tagged lines are functional as 88 of 114 protein traps in (predicted) essential genes are viable. In the same paper they state that only 58 of 63 gene traps (within the same gene set of 114 genes) are actually lethal. Thus, not all gene traps in “essential” genes are lethal and hence the authors' conclusion of 77% is not valid. They should rather only count how many of the 58 lethal gene traps were reverted to viable proteins traps when inserting the GFP cassette. That number should be more accurate and takes potential exon skipping into consideration*.

We recalculated the functionality of tagged proteins using the 58 lethal gene traps as suggested. Of those 58, 42 reverted lethality (72%). This percentage is not significantly different from the 77% that was derived from all 114 essential genes. The manuscript was edited to address these issues.

*More generally, the conclusion that 77% of the tagged proteins “retain function or are not severely affected” is based, at least in part, on the assumption that their EGFP cassette sequence is incorporated into nearly 100% of the mature message. However, sequence context could potentially affect the efficiency of cassette splicing. The authors should consider toning down their language*.

*For example, in the second paragraph of the subsection headed “Knockdown of alpha-Catenin with iGFPi and deGradFP recapitulates known phenotypes”, in the Results section, the authors state: “Hence, we are not able to detect untagged protein in homozygous tagged animals, indicating that the vast majority or all protein is tagged.“ However, a small amount of endogenous protein appears visible on the presented* alpha-Cat*, Dlg1 and Brp westerns. Therefore the authors should rewrite this sentence. Likewise, in the same paragraph, when stating that “westerns with anti-GFP show a nearly identical banding pattern in these flies, demonstrating that little or no exon skipping is occurring (*Figure 4*, bottom), the authors should delete “or no”. Addressing such caveats throughout the paper and in the discussion would in no way diminish the usefulness of this resource*.

We replaced *alpha-Cat* and Dlg1 (in Figures 4 and 5) Western blots probed with anti GFP antibody with blots that were exposed for a longer time. In both cases these anti GFP blots show lower molecular weight bands that are visible in blots probed with anti *alpha-Cat* and anti Dlg1 antibodies. This suggests that these bands are not endogenous untagged protein. However, we cannot rule out the possibility that a low level of exon skipping does occur. Therefore, we have toned down the language of our conclusions and modified the text for these sections.

*2) The beauty of a GFP tagged protein is to investigate its sub cellular localisation. However, the authors only provide tissue expression data and that only from the larval nervous system. This information does not go much beyond the FlyAtlas data of gene expression. Not even the nuclear factors like Jumu-GFP or H6-like homeobox-GFP are shown to localize in the nuclei*.

We have added a new Figure 2 documenting GFP expression in different tissues including muscles, eye disc, salivary gland, ovary, and testis. The subcellular localization (to cytoplasm, nuclei and cell membranes) of some proteins is also shown in Figure 2.

Were any new or unexpected patterns discovered?

We have tagged more than a hundred novel, uncharacterized genes and hence expression patterns for these genes are novel (http://flypush.imgen.bcm.tmc.edu/pscreen/rmce). We cannot state if these patterns were unexpected or not.

*The authors state that Ac78c-GFP labels a subset of neuronal membranes. I cannot see this from the data presented*.

We have removed this figure and related text from the manuscript. However, we now document subcellular localizations in Figure 2 for various genes.

*A few images particularly carefully comparing antibody and tagged-GFP labelling will be useful*.

We presented these data for *alpha-Cat*, Brp, Dlg1, and Dunce. Additionally, we have added two new genes: Dl and EcR in Figure 2—figure supplement 1.

*3)*
Figure 2
*is redundant with*
Figure 2*.*

We have removed the original Figure 2 (original Figure 2 is now Figure 3).

It would be much more useful to give an overview how the 200 genes for expression analysis were selected from the 400 tagged genes, which again were selected from the 1,600 MiMICs. Can the authors expand their explanation for how these lines were chosen that is a bit more specific than “non-random”? Why were they “suspected” to be expressed in the nervous system? Have they been selected using fly atlas above a certain RNA expression level?

The genes were selected because colleagues in the fly community asked us to tag their genes of interest. As many are neuroscientists we expected that many genes would be expressed in the nervous system. This point is now clarified in the manuscript.

*4) The paper mainly focuses on the larval nervous system. As this resource is supposed to be of general use, it would be useful to show that the used muscle myosin heavy chain splice acceptor is indeed efficiently used in other tissues like muscle, epidermis, endoderm etc. This could be easily shown by selecting a couple of ubiquitous essential genes that should result in lethality when GFP RNAi is driven in various tissues*.

As stated in response to reviewer comment #2, we have created a new Figure 2 with examples of GFP expression in a variety of different tissues. We have also shown phenotypes associated with knockdown of these tagged proteins in a variety of tissues types: imaginal discs (*alpha-Cat*, Figure 4; Dlg1, Figure 5), eye (*alpha-Cat*, Figure 4 and Figure 4—figure supplement 2), larval gut (Dlg1, Figure 5—figure supplement 1) and adult brain (Dnc, Figure 9).

*5) The authors nicely show that* α-CatGFP *overlaps with anti-α-Cat antibody in eye discs and that deGradFP and GFP RNAi driven by ey-GAL4 can result in some eye phenotypes in these flies. How do these phenotypes compare to the various available* α-Cat *RNAi lines? How strong is the protein reduction? As western blots of the Cat-GFP lines are shown, these data should be easy to acquire*.

Loss of *alpha-Cat* causes cell lethality, therefore knockdown of *alpha-Cat* in the eye results in a reduction in eye size. Western blotting of this tissue is difficult because of very unequal loading of knockdown samples versus controls. To address how the phenotypes with knockdown using deGradFP and iGRPi compare to the available *alpha-Cat* RNAi lines, we have added a new Figure 4—figure supplement 2 showing eye images after knockdown in eyes. We performed knockdown of *alpha-Cat* in eyes using three different available RNAi lines FBst0033430, FBst0038987, FBst0038197. FBst0033430, causes rough eyes and reduction in eye size that is comparable to deGradFP knockdown of alpha-Cat-EGFP. This is in agreement with the only RNAi knockdown of α-Cat in the eye published by Ulrich Tepass and colleagues (44). The other two, FBst0038987 and FBst0038197, cause pupal lethality with little or no head development. However, no data are available related to the specificity of these RNAis. These results are now discussed in the manuscript.

*It would be even more important to generalize knock-down efficiency by testing a few different number used genes with western blots. Again, these are straight forward experiments that would be very valuable for the community to judge, how well the technology works in general and not only in a few selected examples*.

We have added a new Figure 8—figure supplement 1 with three additional examples. For these experiments, we shifted adult animals raised at 18°C to 28°C for 1-3 days to express deGradFP under the control of either the n-Syb-GAL4 or actin5C-GAL4 driver. Western blots of head extracts against GFP showed a range of knockdown efficiencies from 45% to nearly 100%. The efficiency of protein knockdown, as observed by Western blot, is highly dependent on the specific gene being targeted, the GAL4 driver being used, the developmental stage of the animal, the temperature, the time the animals are kept at 28°C, etc. These parameters were already discussed in the previous version of the manuscript.

*6) The authors argue that one major advantage of their technology is to eliminate gene function tissue specifically*. *Following this logic, why was Tub-GAL4 used to knock-down Dlg1-GFP in*
Figure 4*?*

We used tub-Gal4 to knock down Dlg1-EGFP-Dlg1 in the whole animal in a temperature dependent manner to compare with *dlg1* maternal and zygotic null mutant phenotypes in a variety of tissues (i.e. brain, imaginal disc, gut).

*The loss of epithelial polarity in*
Figure 4
*vs. g is not clear. Better data need to be provided supporting that conclusion*.

We have removed those images from the manuscript. We found it challenging to obtain better images to show distinct polarity because cellular morphology and epithelial organization in the Dlg1-EGFP knockdown animals are severely disrupted. We have therefore, added a new Figure 5—figure supplement 1 that highlights this aberrant cellular organization in larval gut. We have adapted the manuscript to eliminate our claims about disruption of polarity and to highlight the differences in cellular morphology.

*7) We do not understand the interpretation of the*
Figure 5*. anti-Brp clearly shows two bands on western from wild type but only one from Brp-GFP flies. However, anti-GFP detects two bands from Brp-GFP with inverted intensities. The authors interpret these data such that anti Brp does not recognize some Brp-GFP isoforms, because of the GFP insertion. My interpretation is that rather splicing is abnormal, inverting the intensities and partially skipping the exon that contains the anti-Brp epitope*.

We repeated the experiment with an increased amount of sample, longer transfer time and a more sensitive detection system. With these protocol changes, we were able to detect two bands in the Brp-EGFP-Brp blot when probed with anti Brp (nc82) antibody. We assume that, on the original blot, the large band did not transfer properly. We have changed the figure (Figure 6) and the text accordingly.